# Positive selection in the genomes of two Papua New Guinean populations at distinct altitude levels

Mathilde André [1,2], Nicolas Brucato [3], Georgi Hudjasov[2], Vasili Pankratov [2], Danat Yermakovich [2], Francesco Montinaro [1,4], Rita Kreevan [2], Jason Kariwiga[5,6], John Muke[7], Anne Boland [8], Jean-François Deleuze [8], Vincent Meyer[8], Nicholas Evans [9], Murray P. Cox [10,11], Matthew Leavesley [5,12,13], Michael Dannemann [2], Tõnis Org[2], Mait Metspalu [1], Mayukh Mondal [2,14,15] ✉ & François-Xavier Ricaut [3,15] ✉

Highlanders and lowlanders of Papua New Guinea have faced distinct environmental stress, such as hypoxia and environment-specific pathogen exposure, respectively. In this study, we explored the top genomics regions and the candidate driver SNPs for selection in these two populations using newly sequenced whole-genomes of 54 highlanders and 74 lowlanders. We identified two candidate SNPs under selection - one in highlanders, associated with red blood cell traits and another in lowlanders, which is associated with white blood cell count – both potentially influencing the heart rate of Papua New Guineans in opposite directions. We also observed four candidate driver SNPs that exhibit linkage disequilibrium with an introgressed haplotype, highlighting the need to explore the possibility of adaptive introgression within these populations. This study reveals that the signatures of positive selection in highlanders and lowlanders of Papua New Guinea align closely with the challenges they face, which are specific to their environments.

After the first arrival of modern humans in New Guinea around 50 thousand years ago (kya)[1], they rapidly spread across the different environmental niches of the island[2,3]. Since the Holocene (around 11 kya), the population of Papua New Guinea (PNG) has been unevenly distributed, with most of the population living at altitudes between 1600 and 2400 m above sea level (a.s.l.)[4,5]. This population distribution pattern is remarkable considering the challenges PNG highlanders face at this altitude, like the lower oxygen availability to the body[6]. Indeed,

[1]Estonian Biocentre, Institute of Genomics, University of Tartu, Riia 23b, 51010 Tartu, Tartumaa, Estonia. [2]Centre for Genomics, Evolution & Medicine, Institute of Genomics, University of Tartu, Riia 23b, 51010 Tartu, Tartumaa, Estonia. [3]Centre de Recherche sur la Biodiversité et l'Environnement (CRBE), Université de Toulouse, CNRS, IRD, Toulouse INP, Université Toulouse 3 – Paul Sabatier (UT3), Toulouse, France. [4]Department of Biosciences, Biotechnology and the Environment, University of Bari, Bari, Italy. [5]Strand of Anthropology, Sociology and Archaeology, School of Humanities and Social Sciences, University of Papua New Guinea, University 134, PO Box 320 National Capital District, Papua New Guinea. [6]School of Social Science, University of Queensland, St Lucia, QLD, Australia. [7]Social Research Institute Ltd, Port Moresby, Papua New Guinea. [8]Université Paris-Saclay, CEA, Centre National de Recherche en Génomique Humaine (CNRGH), 91057 Evry, France. [9]ARC Centre of Excellence for the Dynamics of Language, Coombs Building, Fellows Road, CHL, CAP, Australian National University, Canberra, ACT, Australia. [10]School of Natural Sciences, Massey University, Palmerston North, New Zealand. [11]Department of Statistics, University of Auckland, Auckland, New Zealand. [12]College of Arts, Society and Education, James Cook University, P.O. Box 6811 Cairns, QLD 4870, Australia. [13]ARC Centre of Excellence for Australian Biodiversity and Heritage, University of Wollongong, Wollongong, NSW 2522, Australia. [14]Institute of Clinical Molecular Biology, Christian-Albrechts-Universität zu Kiel, 24118 Kiel, Germany. [15]These authors jointly supervised this work: Mayukh Mondal, François-Xavier Ricaut. ✉e-mail: mondal.mayukh@gmail.com; francois-xavier.ricaut@univ-tlse3.fr

various detrimental conditions, such as reduced birth weight[7] and shorter life span[8], have been observed at altitudes as low as 1500 m a.s.l. Studies investigating the hypoxic response of the human body in high-altitude populations (living above 2500 m a.s.l.) revealed that selection acted on genes involved in the Hypoxia-Inducible Factor (HIF) pathway[9], which is the principal response mechanism to low oxygen at the cellular level. This pathway regulates angiogenesis, erythropoiesis, and glycolysis[10]. Some high-altitude populations show a limited increase in haemoglobin concentration[11] in response to the lower oxygen levels. Indeed, an increase in haemoglobin concentration – as observed in native lowlanders ascending to altitude – boosts oxygen transport but also results in higher blood viscosity[12]. In the long term, that process may cause Chronic Mountain Sickness (CMS) and cardiovascular complications[12]. Interestingly, Tibetan highlanders show selection that is associated with a more restrained increase of haemoglobin concentration at altitude due to increased plasma volume[13]. This suggests that hypoxia might lead to the selection of a complex haematological response that overcomes the increase in blood viscosity when enhancing oxygen transport. Signatures of selection have also been observed in populations living at intermediate altitudes (above 1500 m a.s.l.)[14,15]. For example, the Andean Calchaquíes carry genomic signatures of selection for pathways associated with the nitric oxide metabolism and with the neurotransmitter GABA[16]. In addition, signatures of positive selection to altitude have also been found among Ethiopians currently living at 1800 m a.s.l.[15] and in the Caucasus population living at intermediate altitudes of 2000 m a.s.l.[14]. These studies suggest that the genomic signature of selection can occur even at intermediate altitudes in response to more moderate selection pressure.

However, the role of selection in response to the environmental challenges by altitude on the genomes of PNG highlanders, who inhabited this environment for the last 20,000 years[3], remains mostly unknown. PNG highlanders significantly differ from PNG lowlanders in height, chest depth, haemoglobin concentration, and pulmonary capacities[17]. Similar differences have been observed between Andean, Tibetan and Ethiopian highlanders and their corresponding lowland populations[18]. However, various factors, like phenotypic plasticity[19], diet or physical activities, could explain these phenotype differences. In this paper, we explored whether these phenotypes can also be linked to adaptive processes acting on the genome of the PNG highlanders.

Another strong environmental pressure in PNG is infectious diseases (e.g., malaria, dysentery, pneumonia, tuberculosis, etc) that are the leading cause of death in PNG[20]. In this pathogenic environment, malaria stands out among others because it might affect highlanders and lowlanders differently. The incidence of malaria varies enormously between the lowlands and the highlands. While PNG accounted for nearly 86% of the malaria cases in the Western Pacific Region in 2020[21], malaria is practically absent in PNG highlands, possibly because of a limited dispersal of *Anopheles*, the main vector of malaria, at high altitude[4]. It has been suggested that malaria might explain the unbalanced population distribution between PNG highlands and lowlands[22] and thus induces a selection pressure specific to lowlanders. Nonetheless, the period when this specific pathogenic pressure started to impact Papuans remains unclear[22].

Besides facing these environmental pressures, PNG populations also stand out by their high levels of Denisovan introgression[23]. Denisovan introgressed variant might contribute to Tibetans' adaptation to altitude[15] and affect the immune system of the PNG population[24]. Moreover, because some archaic variants show signals of selection among the overall Papuan population[25–27], it is conceivable that archaic introgression has contributed to beneficial alleles in PNG populations. However, to date, it remains elusive to which extent archaic introgression contribution to local adaptation varies between PNG populations.

In this study, we identified the genomic regions that show signatures of selection in 54 newly sequenced PNG highlanders and 74 lowlanders. We then screened for the SNP that most likely drives the selection signal in each genomic region under selection. We also explored phenotype associations with candidate SNPs. Finally, we scanned regions under selection for the presence of introgressed archaic haplotypes and assessed the role of introgressed alleles on adaptive processes. Our research provides new insights into local adaptation in PNG populations and its implications on health.

## Results

### Selection scans results in PNG highlanders and PNG lowlanders

To study selection specific to PNG highlanders or PNG lowlanders, we used 54 newly sequenced genomes from three villages in PNG Highlands located in Mount Wilhelm between 2300 and 2700 m above sea level (a.s.l.) and 74 newly sequenced genomes of PNG lowlanders from Daru Island (<100 m a.s.l.). PCA, ADMIXTURE and D statistic results support that these two populations are homogeneous and show limited level of admixture from outside PNG, suggesting that the recent gene flows from populations originating from Asia and Europe would have a minor impact on the selection scan (Supplementary Figs. 3 and 4, Supplementary Table 3). An important consideration in our study design is the proximity of the PNG highlander and PNG lowlander populations. ADMIXTURE analysis revealed that PNG lowlanders exhibited an average of 3.23% admixture (SD = 5.29%) from PNG highlanders, indicating potential historical gene flow between these populations. This genetic exchange might introduce signals of selective sweeps that did not originate in the target population[28]. However, it's worth noting that we employed the source populations of the admixture as reference populations for our selective sweep analysis (Supplementary Note 11). As shown in Supplementary Figs. 10 and 11, this approach effectively mitigates the potential impact of such genetic exchange on our selection scans results. Moreover, we show that the PNG lowlanders with higher coverage have a limited impact on our selection scan results (Supplementary Note 10, Supplementary Tables 8–10).

We computed frequency-based (PBS) and haplotype-based (XP-EHH) selection statistics – two selection tests based on distinct genetic signatures – to detect candidate regions for selection in PNG highlanders and lowlanders. Both selection statistics require a target and reference population, allowing us to identify the signal of selection within the target population (PNG highlanders or PNG lowlanders) but absent in the reference population (PNG lowlanders or PNG highlanders, respectively). We also combined both these statistics in a Fisher Score[25] to detect the region with extended haplotype homozygosity and carrying multiple variants with high allele frequency. We kept the ten regions with the highest score and p-value below $2 \times 10^{-5}$, $2 \times 10^{-4}$ or $2 \times 10^{-3}$ for XP-EHH, PBS and Fisher Score, respectively, leading to 30 genomic regions of interest for PNG highlanders and lowlanders (Supplementary Figs. 5–7, Supplementary Tables 4 and 5). We merged the overlapping regions between methods, resulting in a final number of 21 regions of interest in PNG highlanders (Table 1, Fig. 1) and 23 in PNG lowlanders (Table 2, Fig. 2).

The 21 regions showing signatures of selection in PNG highlanders encompass 54 genes, including genes involved in the regulation of platelet adhesion (ex: *FBLN1*[29]), HIF pathway (ex: *LINC02388*[30]), neurodevelopment (ex: *DLGAP1*[31]) and immunity (ex: MHC locus[32]) (Table 1, Fig. 1). The region with the highest Fisher score and second highest PBS and XP-EHH scores in PNG highlanders includes the long intergenic non-protein coding RNA *LINC02388*. This intergenic RNA is associated with the serum levels of protein LRIG3[30] that impact angiogenesis – the formation of new blood vessels – in glioma cells through regulation of the HIF-1α/VEGF pathway[33]. Comparably to other axes of the HIF pathway under selection in high-altitude populations[9], we hypothesize that this selection signature on

## Table 1 | Merged regions under selection and SNPs most likely to be selected in PNG highlanders

| Merged top regions | Score | Protein coding genes in the region | Candidate SNP for the region | DAF | Significant association (UK Biobank) | Introgressed haplotype in PNG highlanders | Archaic origin |
|---|---|---|---|---|---|---|---|
| chr1:95529290-95736826 | XPEHH | – | rs887476833-G>A | 0.55 | –[b] | – | – |
| chr2:151012094-151201575 | PBS | – | rs74621527-G>A | 0.92 | – | **chr2:151077551-151194524** | Neanderthal |
| chr3:13010340-13217789 | XPEHH | *IQSEC1* | rs374181005-C>T | 0.41 | –[b] | chr3:13132090-13174330 | Denisovan |
| chr3:61779523-62009858 | PBS, Fisher | *PTPRG* | rs79600167-G>A | 0.77 | – | chr3:61798966-61853037 | Neanderthal |
| chr4:110182324-110384099 | XPEHH | *ELOVL6* | rs943845085-A>G | 0.42 | –[b] | chr4:110232325-110334098 | Neanderthal |
| chr4:152704503-152970509 | XPEHH | TIGD4, ARFIP1, *FHDC1* | rs369030953-A>G | 0.59 | – | – | – |
| chr6:30916070-31153184[a] | XPEHH | VARS2, SFTA2, MUCL3, MUC21, **MUC22**, HCG22, C6orf5, PSORS1C1, CDSN, PSORS1C2, PSORS1C1, CCHCR1 | rs940110341-A>C | 0.61 | – | chr6:31077777-31112941 | ambiguous |
| chr6:33006055-33132312[a] | PBS, Fisher | HLA-DAO, HLA-DPA1, **HLA-DPB2** | rs9277772-T>C | 0.21 | Body proportion, blood composition, other phenotypes (Supplementary Table S13) | – | – |
| chr7:147590904-147718219 | PBS | *CNTNAP2* | rs17170618-T>C | 0.52 | – | chr7:147665094-147696027 | Denisovan |
| chr9:85458922-85745092 | XPEHH | *AGTPBP1* | rs28728004-C>A | 0.69 | – | – | – |
| chr10:131112245-131235951 | PBS | *TCERG1L* | rs10299909-T>G | 0.43 | – | chr10:131130857-131157433 | Denisovan |
| chr12:6452552-6662260 | XPEHH | LINC02388[c], TAPBPL, VAMP1, MRPL51, GAPDH, NOP2, LPAR5, ING4, ACRBP, CHD4,IFFO1, NCAPD2 | rs74576183-A>G | 0.71 | Blood composition (Table S13) | – | – |
| chr12:9886812-10055333 | Fisher | KLRF2, CLEC2A, **CLEC12A**, CLEC1B, CLEC12B, CLEC9A | rs536947-C>T | 0.91 | – | chr12:9904201-10023903 | Denisovan |
| chr12:58391529-58634980 | XPEHH, PBS, Fisher | – | rs376870800-C>T | 0.70 | – | **chr12:58451248-58568114** | ambiguous |
| chr12:103783315-104121479 | Fisher | **NT5DC3**, HSP90B1, GLT8D2, HCFC2, NFYB, TDG | rs1032698711-G>A | 0.47 | –[b] | chr12:103839272-104061448 | Denisovan |
| chr13:47639988-47825193 | PBS | – | rs1033760372-C>A | 0.19 | –[b] | – | – |
| chr13:104734734-104875020 | PBS, Fisher | – | rs16965509-G>A | 0.50 | - | chr13:104787393-104824094 | Denisovan |
| chr14:60157772-60377317 | Fisher | PCNX4, **DHRS7**, PPM1A | rs1033848215-A>G | 0.32 | – | – | – |
| chr14:92230479-92401520 | Fisher | *SLC24A4* | rs8003454-C>T | 0.52 | – | chr14:92370144-92392663 | ambiguous |
| chr18:4072997-4251153 | XPEHH, Fisher | *DLGAP1* | rs371858795-T>C | 0.77 | – | chr18:4136427-4203633 | Denisovan |
| chr22:45519818-45644906 | PBS, Fisher | *FBLN1* | rs1601558750-C>T | 0.10 | – | – | – |

Genomic coordinates are given for GRCh38.

Genes in bold are the closest to the candidate SNP defined with CLUES for the region.

The introgressed archaic haplotypes with the highest frequency in each candidate region for selection in PNG highlanders are reported. Introgressed haplotype with which the candidate SNP in high LD ($r^2 > 0.5$) with at least one archaic SNP are in bold. The putative source of introgression is based on hmmix results.

DAF is given for PNG highlanders.

[a]Reference Assembly Alternate Haplotype Sequence Alignments.

[b]Candidate SNP was not present in the UK Biobank, association is shown for the closest SNP within 50 bp upstream and downstream region.

[c]long intergenic non-protein coding RNA.

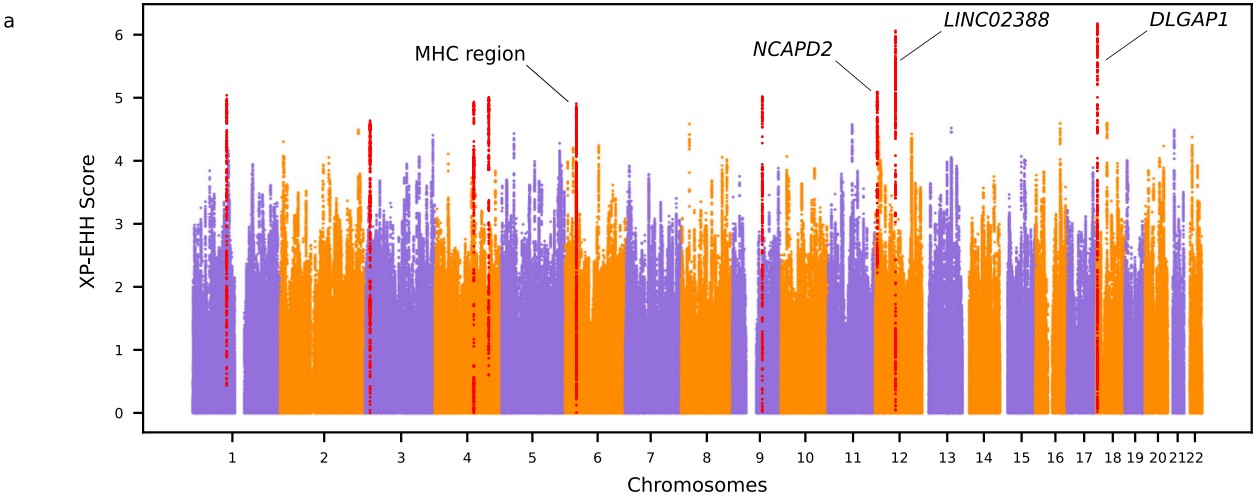

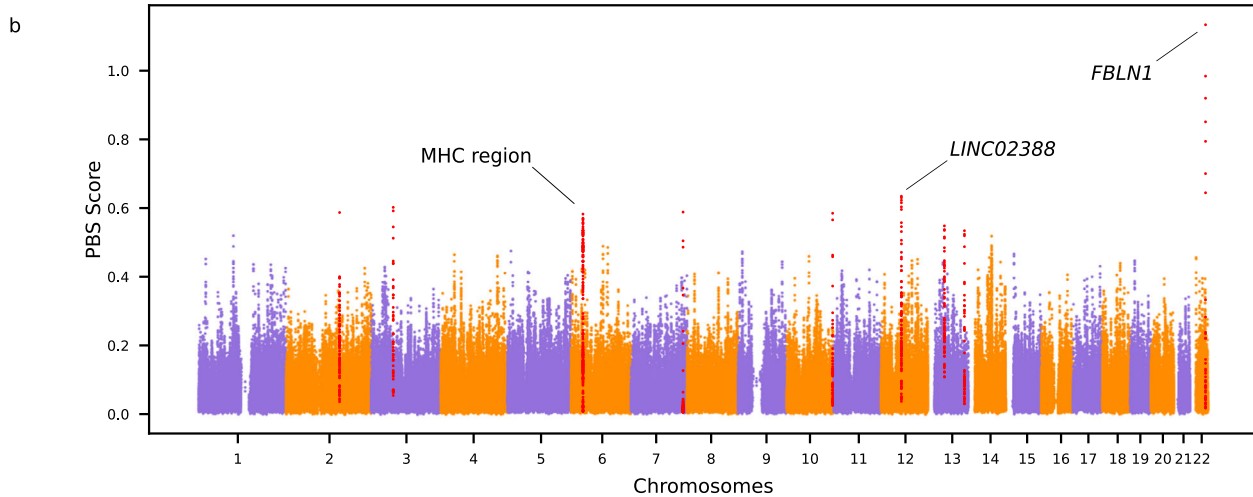

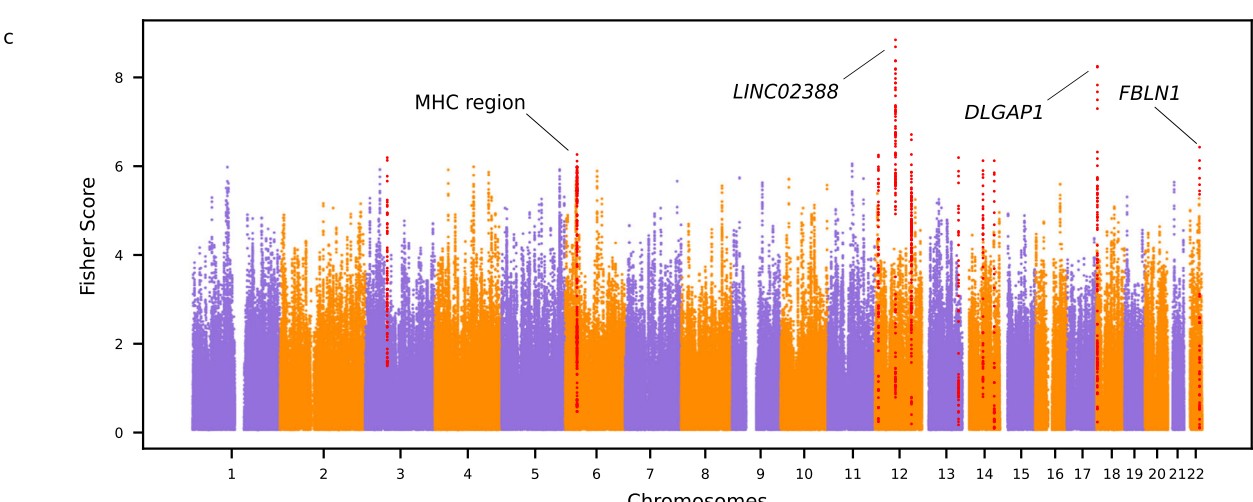

**Fig. 1 | Manhattan plots for the three selection scans among PNG highlanders.** The top ten genomic regions with the highest score are shown in red. Candidate genes discussed in the paper are marked. **a** XP-EHH scores using PNG highlanders as the target population and PNG lowlanders as the reference population. **b** PBS scores using PNG highlanders as the target population, PNG lowlanders as the reference population, and Yorubas from 1000G as the outgroup. **c** Fisher Scores combining the PBS and XP-EHH scores of PNG highlanders. Source data are provided as a Source Data file.

**Table 2 | Merged regions under selection and SNPs most likely to be selected in PNG lowlanders**

| Merged top regions | Score | Protein coding genes in the region | Candidate SNP for the region | DAF | Significant association (UK Biobank) | Introgressed haplotype | Archaic origin |
|---|---|---|---|---|---|---|---|
| chr1:88800562-89326878 | XPEHH, PBS, Fisher | PKN2, GTF2B, KYAT3, RBMXL1, GBP3, GBP1, **GBP2**, GBP7, GBP4, GBP5 | rs368120563-T>C | 0.87 | – | **chr1:89054418-89202534** | ambiguous |
| chr1:237827847-237992467 | PBS | RYR2, **ZP4** | rs157415437-T>C | 0.14 | –[b] | – | – |
| chr1:124085628-124249405 | PBS | **CNTNAP5** | rs7583123-G>T | 0.49 | – | – | – |
| chr2:200238798-200432145 | PBS | **SPATS2L** | chr2:200269472-A>G | 0.05 | –[b] | – | – |
| chr2:241759136-242088831[a] | XPEHH, PBS, Fisher | GAL3ST2, NEU4, **PDCD1**, RTP5, FAM240C | rs376150658-C>G | 0.23 | –[b] | chr2:241811883-241869518 | Neanderthal |
| chr4:82750503-83146792 | Fisher | SCD5, **SEC31A**, LIN54, COPS4, PLAC8 | rs4693058-C>T | 0.76 | Blood composition | chr4:82755644-83083169 | Denisovan |
| chr4:171791098-171986729 | Fisher | **GALNTL6** | rs926184421-G>T | 0.08 | Other phenotypes[b] | – | – |
| chr5:65504470-65708617 | XPEHH | **CENPK**, TRIM23, SGTB, PPWD1, SHLD3, TRAPPC13 | rs36003688-T>C | 0.31 | – | – | – |
| chr6:85266477-85483888 | PBS | **NT5E** | rs989789809-T>C | 0.14 | –[b] | chr6:85340299-85364688 | Denisovan |
| chr7:129548370-129836070 | XPEHH, Fisher | **NRF1**, UBE2H | rs6950082-T>A | 0.49 | Blood composition, other phenotypes[b] | chr7:129553314-129774681 | Denisovan |
| chr8:133791891-133961825 | PBS | – | rs187915256-A>G | 0.99 | –[b] | – | – |
| chr9:93717217-93877803 | XPEHH | – | rs372277219-G>A | 0.22 | – | chr9:93752325-93867864 | Neanderthal |
| chr12:120353731-120666335 | Fisher | MSI1, **COX6A1**, GATC, TRIAP1, SRSF9, DYNLL1,COQ5, RNF10,POP5, CABP1 | rs75047318-T>C | 0.07 | Blood composition, body proportion, respiratory capacities, other phenotypes | chr12:120368947-120395906 | ambiguous |
| chr13:61590770-61993327 | XPEHH | – | rs537391125-A>G | 0.94 | –[b] | – | – |
| chr13:89660867-89920623[a] | Fisher | – | rs72634302-G>A | 0.48 | – | – | – |
| chr14:37137933-37382802 | XPEHH | SLC25A21, **MIPOL1** | rs1594377001-G>A | 0.05 | –[b] | – | – |
| chr14:77312867-77558267 | PBS, Fisher | POMT2, GSTZ1, SAMD15, NOXRED1, VIPAS39, ISM2,**SPTLC2**, TMED8, AHSA1 | rs12885954-C>T | 0.57 | – | – | – |
| chr16:87806834-87928392 | XPEHH | **SLC7A5**, CA5A | rs2287123-G>A | 0.32 | Other phenotypes | – | – |
| chr17:54003406-54222843 | XPEHH | – | rs575590765-G>A | 0.11 | –[b] | chr17:54036011-54160414 | Denisovan |
| chr18:41133289-41618597 | Fisher | – | rs2848745-G>C | 0.95 | – | – | – |
| chr19:11708670-12108034 | PBS | ZNF823, ZNF441, ZNF491, ZNF440, ZNF439, ZNF69,ZNF700, ZNF763, ZNF433, ZNF20, ZNF878, **ZNF844** | rs900717974-C>T | 0.11 | –[b] | chr19:11708670-12108034 | Neanderthal |
| chr19:16344294-16576199 | XPEHH | **EPS15L1**, CALR3, CHERP, C19orf44, SLC35E1, MED26 | rs1870071-C>T | 0.76 | Blood composition | – | – |
| chr19:54176104-54330609[a] | PBS, Fisher | MBOAT7, TSEN34, **RPS9**, LILRB3, LILRA6, LILRB5, LILRB2, LILRA5 | rs1600734199-T>C | 0.13 | –[b] | – | – |

Genomic coordinates are given for GRCh38.
Genes in bold are the closest to the candidate SNP defined with CLUES for the region.
The introgressed archaic haplotypes with the highest frequency in each candidate region for selection in PNG lowlanders are reported. Introgressed haplotype with which the candidate SNP in high LD ($r^2 > 0.5$) with at least one archaic SNP are in bold. The putative source of introgression is based on hmmix results.
DAF is given for PNG lowlanders.
[a]Reference Assembly Alternate Haplotype Sequence Alignments.
[b]Candidate SNP was not present in the UK Biobank, association is shown for the closest SNP within 50 bp upstream and downstream region.

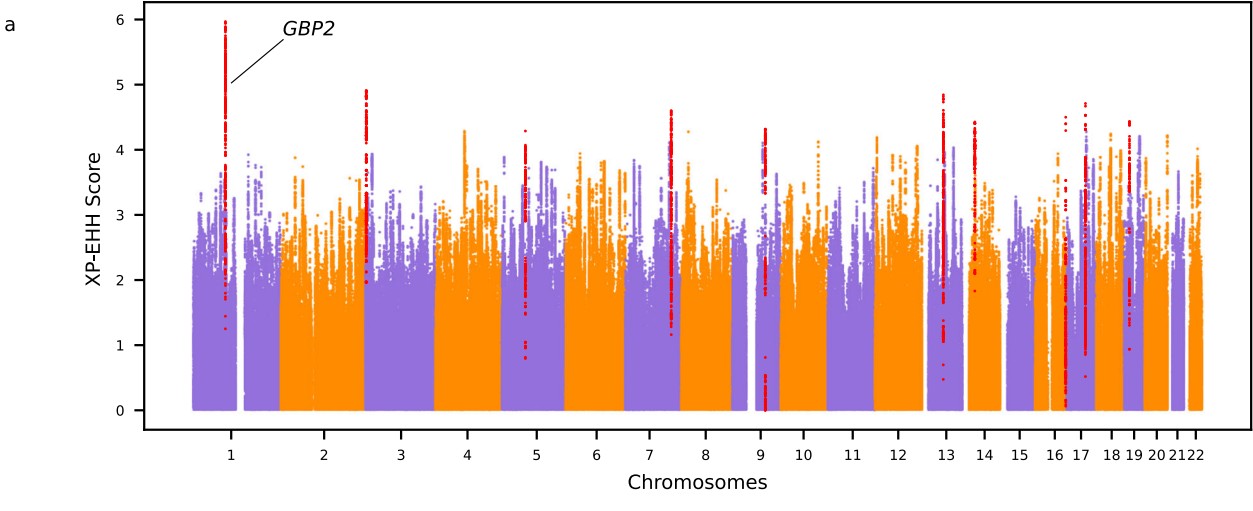

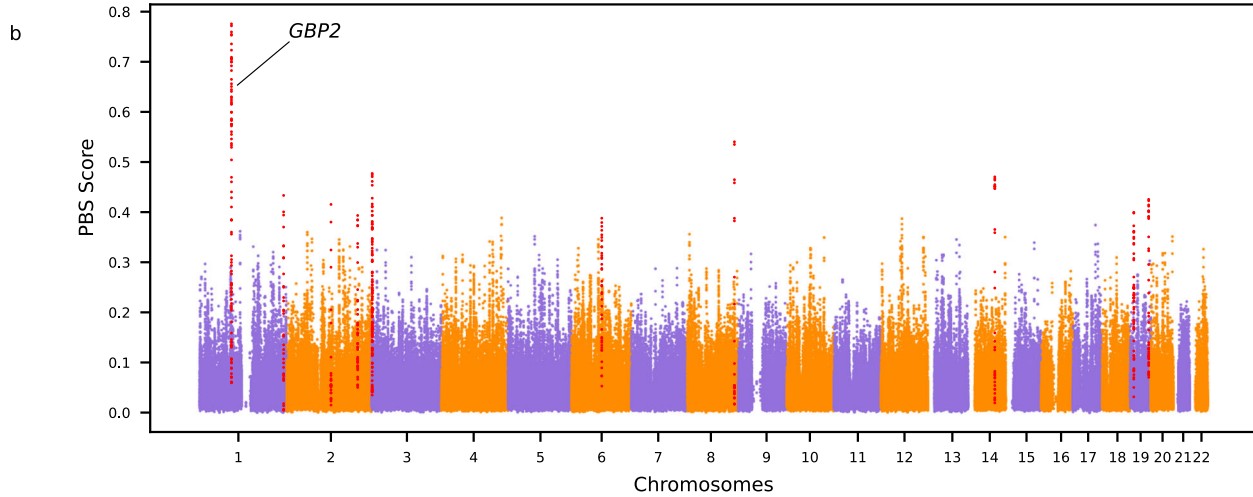

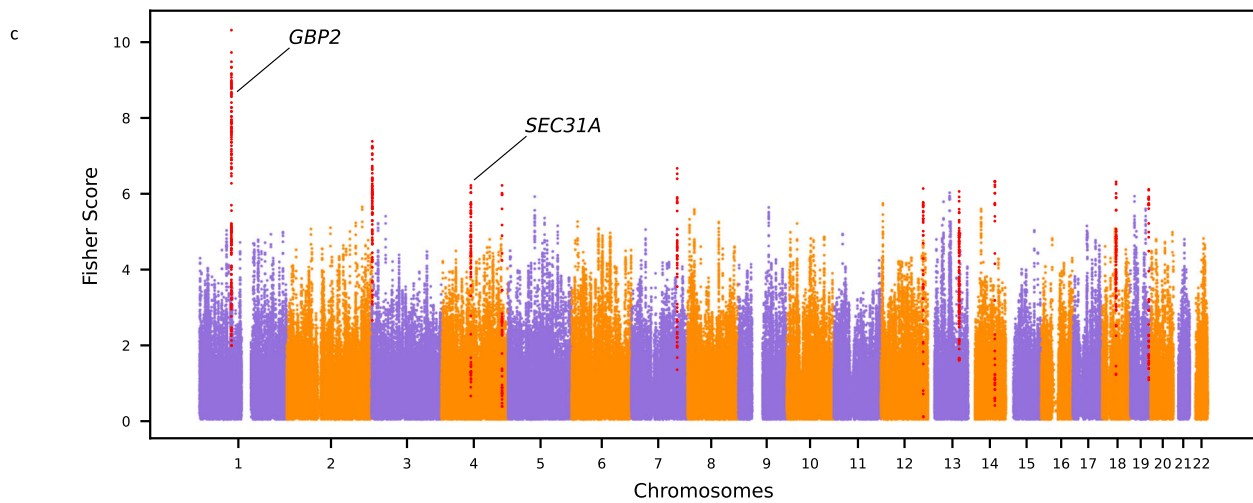

**Fig. 2 | Manhattan plots for the three selection scans among PNG lowlanders.**
The top ten genomic regions with the highest score are shown in red. Candidate
genes discussed in the paper are marked. **a** XP-EHH scores using PNG lowlanders as
the target population and PNG highlanders as the reference population. **b** PBS

scores using PNG lowlanders as the target population, PNG highlanders as the
reference population, and Yorubas from 1000G as the outgroup. **c** Fisher Scores
combining the PBS and XP-EHH scores of PNG lowlanders. Source data are pro-
vided as a Source Data file.

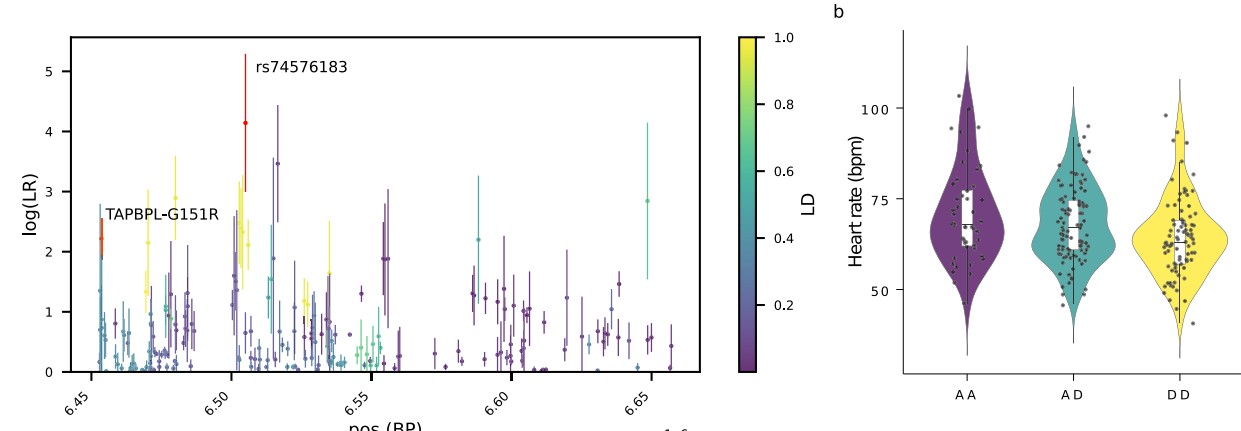

**Fig. 3 | Clues logLR and violin plots of the heart rate distribution depending on the genotype of the candidate SNP for the regions chr12:6452552-6662260 under selection in PNG highlanders. a** LogLR for SNPs in regions under selection after five runs of CLUES or 50 runs of CLUES for each of the top five SNPs in the candidate region. The candidate SNP rs74576183-A>G driving selection for the region is shown in red. The missense variant (rs7295376, TAPBPL-G151R) in high LD with rs74576183-A>G is shown in orange. The colour scale indicates linkage disequilibrium with the candidate SNP. CLUES logLR are presented as mean values +/− SD for $n = 5$ independent runs of CLUES (or $n = 50$ for the top five SNPs). **b** Violin plots of the heart rate distribution in PNG individuals (PNG highlanders, PNG lowlanders and PNG diversity set I, $n = 232$) depending on their genotype for the candidate SNP rs74576183-A>G (A = ancestral allele, D = derived allele under selection, AA = AA, AD = AG, DD = GG). The centre of the box-plot represents the median heart rate. The bounds of the box encompass the interquartile range (IQR = Q1–Q3), while the whiskers extend up to 1.5 × IQR units beyond the box boundaries. We used the ggplot2 package (v3.4.2) to generate the violin plots. Source data are provided as a Source Data file.

*LINC02388* might reflect adaptive processes counteracting hypoxia by affecting the formation of new blood vessels. This axis of the HIF pathway might maintain oxygen transport to appropriate levels in PNG highlanders while limiting the increase in haemoglobin concentration and blood viscosity.

Genomic selection candidate regions in PNG lowlanders encompass multiple immunity-related genes (Table 2, Fig. 2). Notably, the region with the highest XP-EHH, PBS and Fisher Score includes several genes from the guanine-binding protein family (GBP). This gene family is associated with protective effects against diverse pathogens[34]. The lowlander-specific selection signature for this gene family might indicate that adaptive processes in this population were linked to the specific pathogenic pressure PNG lowlanders faced.

**Selected SNPs phenotypic associations**
Next, we sought to identify the most likely selection target SNP in each candidate region. To this end, we reconstructed allele frequency trajectories through time for all SNPs in a candidate region for selection for the last 980 generations (27,440 years), using CLUES[35] and picked the SNP with the largest average logLR (here onwards, they will be regarded as candidate SNPs; Tables 1 and 2, Supplementary Tables 6–9). We then applied two complementary approaches to explore the phenotypic effects of each candidate SNP. First, we queried GWAS summary statistics from the UK Biobank for each candidate SNP. Two candidate SNPs of PNG highlanders demonstrate significant association with at least one phenotype of the UK Biobank (Table 1, Supplementary Table 10). Two of these SNPs are significantly associated with haematological phenotypes, which is significantly more than expected under random chance (pval = 0.022). Similarly, among PNG lowlanders, the candidate SNPs - or the closest SNPs in a window of 100 bp when the candidate SNP was not present in the UK Biobank - show significant associations in the UK Biobank and four with haematological phenotypes, which is also a higher number of associations with a haematological phenotype than expected under random chance (pval = 0.038) (Table 2, Supplementary Tables 11 and 12).

When looking for association between the candidate SNPs and the phenotype measurements done for PNG highlanders, PNG lowlanders and PNG diversity set I datasets, after correction for age, gender and the number of tested SNPs, we identified two significantly associated

SNPs, both of which showed associations with heart rate (pval$_{adjusted\_snp}$ < 0.05; pval adjusted for the number of SNPs tested) (Figs. 3 and 4) although this association does not survive after correcting the significance threshold for the number of tested phenotypes groups (pval$_{adjusted\_snp}$ > 0.05/5) (Supplementary Note 15, Supplementary Table 13). The derived allele G of rs74576183-A>G, an intronic variant of *NCAPD2* that is under positive selection in PNG highlanders based on CLUES results (Supplementary Table 6), might be associated with a slower heart rate (pval$_{adjusted\_snp}$ = 0.046, beta = −2.981; Supplementary Table 13, Fig. 3a, b). On the contrary, the derived allele T of rs4693058-C>T, an intronic variant of *SEC31A*, that is under positive selection in PNG lowlanders (Supplementary Table 7) might be associated with a faster heart rate (pval$_{adjusted\_snp}$ = 0.046, beta = 3.137; Supplementary Table 13, Fig. 4a, b). Interestingly, both of these SNPs also exhibited significant associations with haematological phenotypes in the UK Biobank (Supplementary Tables 10 and 11). Specifically, the SNP rs74576183-A>G, which is under selection in PNG highlanders, displayed its most robust association with red blood cell count. In contrast, the SNP rs4693058-C>T, under selection in PNG lowlanders, demonstrated its strongest association with lymphocyte percentage. The association of these two candidate SNPs with heart rate may reflect broader connections with other haematological components, such as red and white blood cell counts, which were not directly measured in the PNG samples. This hypothesis aligns with the fact that multiple haematological factors influencing heart rate are often overlooked and might represent the actual targets of selection[13].

However, both the association approaches mentioned above have limitations. First, associations from the UK Biobank have been detected in a population different from Papua New Guineans; the transferability of the beta values (including the directions) of the associations is therefore limited[36]. Because of this limitation, we avoided using the UK Biobank summary statistics to make any assumptions on the direction of the phenotype association in PNG populations. However, most of the GWAS significant associations are due to common variants shared between populations or variants that map close to the associated SNPs. High replicability of GWAS results has notably been observed between Europeans and East Asians[37], which suggests that the associations we observed between the candidate SNPs and the phenotypes from the UK Biobank most likely remain a relevant proxy

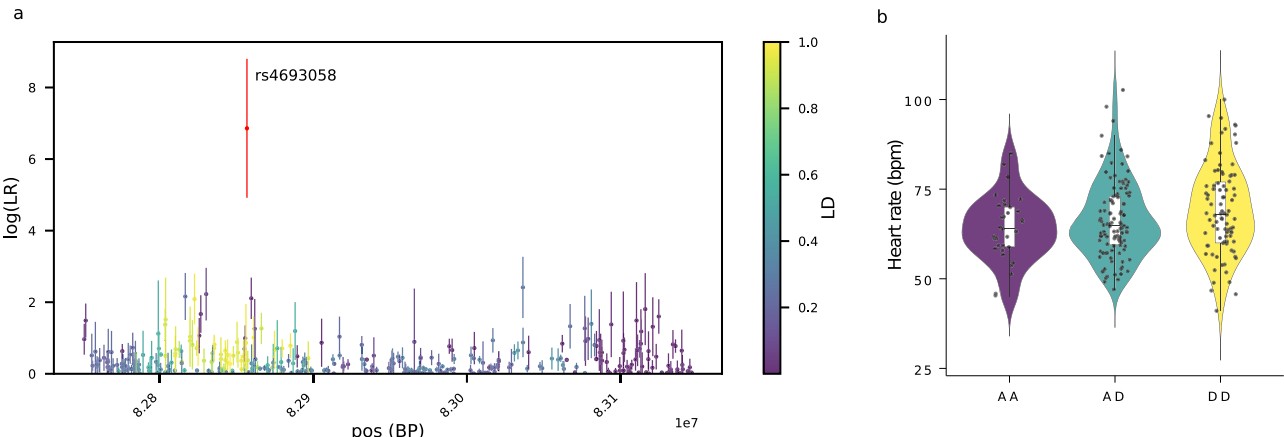

**Fig. 4 | Clues logLR and violin plots of the heart rate distribution depending on the genotype of the candidate SNP for the regions chr4:82750503-83146792 under selection in PNG lowlanders. a** logLR for SNPs in regions under selection after five runs of CLUES or 50 runs of CLUES for each of the five top SNPs in the candidate region. The candidate SNP rs4693058-C>T driving selection for the region is shown in red. The colour scale indicates linkage disequilibrium with the candidate SNP. CLUES logLR are presented as mean values +/− SD for $n = 5$ independent runs of CLUES (or $n = 50$ for the five top SNPs). **b** Violin plots of the heart rate distribution in PNG individuals (PNG highlanders, PNG lowlanders and PNG diversity set I, $n = 232$) depending on their genotype for the candidate SNP rs4693058-C>T (A = ancestral allele, D = derived allele under selection, AA = CC, AD = CT, DD = TT). The centre of the box-plot represents the median heart rate. The bounds of the box encompass the interquartile range (IQR = Q1–Q3), while the whiskers extend up to $1.5 \times$ IQR units beyond the box boundaries. We used the ggplot2 package (v3.4.2) to generate the violin plots. Source data are provided as a Source Data file.

among the PNG populations until a proper biobank based on Papuan ancestry is available.

Secondly, we found no significant phenotype association for candidate SNPs when correcting for the number of SNPs and phenotypes tested together. That may be because of the low sample size or the choice of documented phenotypes that are not the direct target of selection. Nonetheless, finding association with related phenotypes in both analyses supports the hypothesis that cardiovascular phenotypes were a target of selection within PNG highlanders and lowlanders.

### Functional consequences of candidate SNPs

In order to study the potential molecular effects and the most likely target genes of the selection candidate SNPs, we investigated their putative regulatory role and impact on the protein structure. Significant eQTLs for the candidate SNPs are included in Supplementary Tables 14 and 15, and the VEP results for the SNPs in LD with the candidate SNPs ($r^2 \geq 0.5$) can be found in Supplementary Tables 16 and 17. Moreover, we plotted the chromatin state of the candidate SNPs for the epigenomes of the Roadmap project (Supplementary Figs. 13 and 14).

In addition, we scanned the top selected genomic regions for missense variants (Supplementary Tables 18 and 19). In PNG highlanders, one of the regions under selection (chr12:6502552-6612260) overlaps with one missense variant (rs7295376, TAPBPL-G151R) that shows an exceptionally high derived allele frequency (DAF) in PNG highlanders than any other population (DAF = 0.70 vs <0.12 in African, Asian or European populations; Supplementary Table 18). Moreover, this missense variant is in high LD ($r^2 = 0.95$) with the candidate SNP, rs74576183-A>G.

In the case of genomic regions under selection in PNG lowlanders, the selection candidate region encompassing GBP overlaps with a missense variant (rs143126710, GBP2-A549T), which is absent in non-Papuan populations and has a DAF of 0.82 in PNG lowlanders (Supplementary Table 19). This missense variant is part of the top five SNPs given by CLUES for the region (Supplementary Table 9). That might suggest that we failed to identify the real selection driving SNP when limiting the candidate SNPs to the first top one. This variant is in moderate LD ($r^2 = 0.57$) with the candidate SNP for the region

(rs368120563-T>C). While we expect CLUES top results to be enriched for the causal SNPs of selection, it remains possible that the real targets of selection (at least in those cases) are SNPs in high LD with the candidate SNPs. In the case of rs368120563-T>C, under selection in PNG lowlanders, we suggest that the linked missense variant GBP2-A549T modifying protein sequence might be the real target of selection for the genomic region.

### Archaic introgression in loci under selection

We reported the highest frequency archaic haplotype overlapping each top genomic region under selection in PNG highlanders or lowlanders with a putative source of introgression from Altai Neanderthal and Denisovan based on hmmix[38] (Supplementary Tables 20 and 21, Supplementary Figs. 15–18, Supplementary Data 4–5). The region with the highest XP-EHH, PBS and Fisher score in PNG highlanders and carrying *LINCO2388* – that might regulate angiogenesis through the HIF/VEGF pathway – carries an ambiguous archaic introgressed haplotype with archaic SNPs from both Altai Neanderthal and Denisovan (Table 1, Supplementary Table 20, Supplementary Data 4). Within regions under selection in PNG lowlanders that show archaic introgression (Table 2, Supplementary Table 21), the region encompassing the immunity-related GBP locus, which exhibits the highest selection peak in PNG lowlanders, shows haplotypes with sequence similarities to both Denisovan and Altai Neanderthal (Fig. 5, Supplementary Fig. 15, Supplementary Table 21, Supplementary Data 5). Archaic introgression in this region has previously been reported in Melanesians[27,39]. Interestingly, we observed a very low distance difference between Altai Neanderthal and Denisovan for the genomic region that is introgressed in PNG populations (Supplementary Table 22). This supports a shared ancestry between Denisovan and the Altai Neanderthal for that genomic region (interestingly absent in the other two high-coverage Neanderthal) and that we most likely observed Denisovan introgression within the GBP locus in the PNG population. Finally, the introgressed haplotype, including the candidate SNP for selection in the GBP region, has a higher frequency in PNG lowlanders (0.87) than in PNG highlanders (0.57). Future studies are needed to test the significance of archaic introgression contribution to selection signatures in PNG populations.

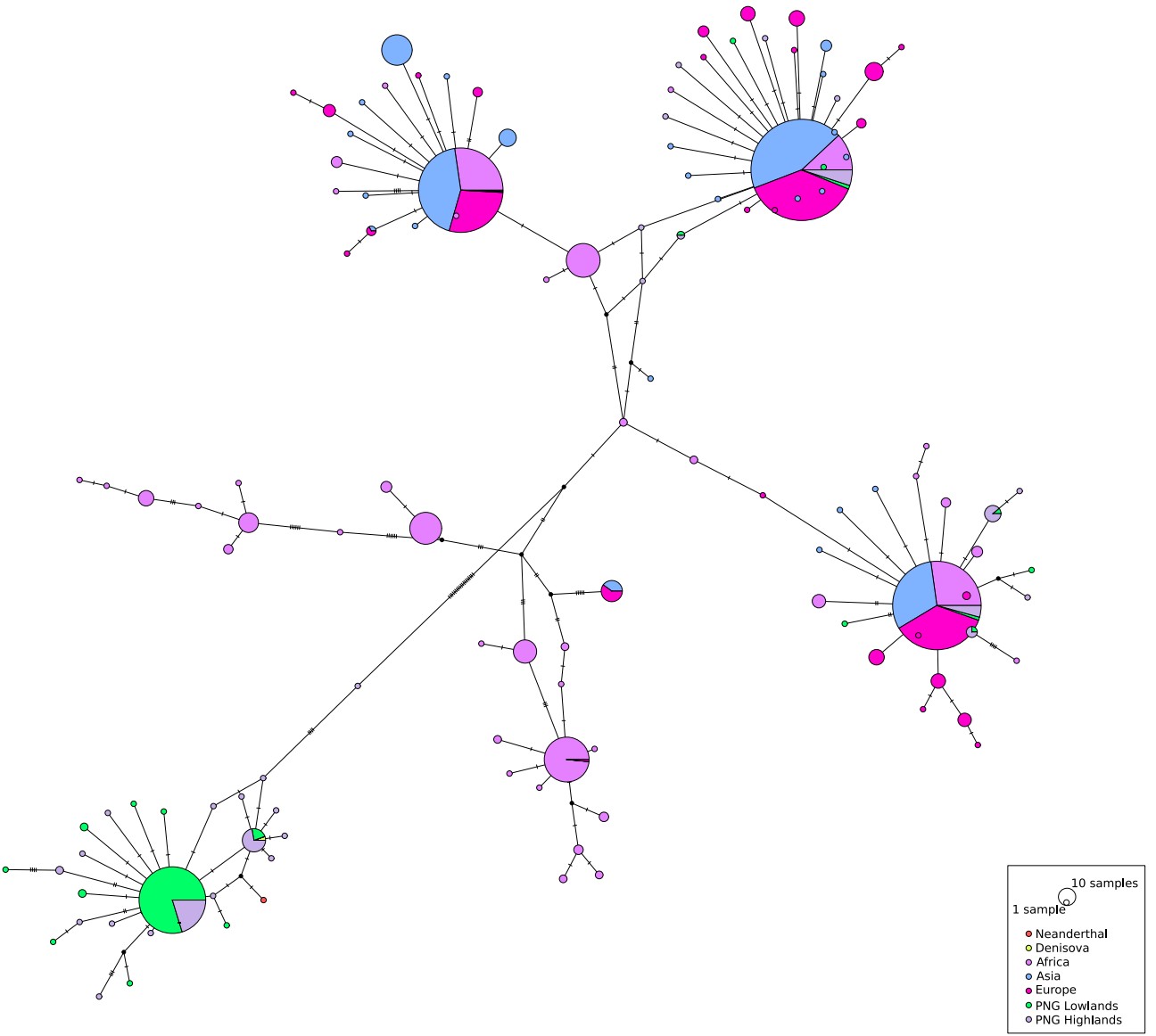

**Fig. 5 | Median-joining haplotype networks for the windows 5kbp down- and upstream rs368120563, the candidate SNP for the genomic region chr1:88800562-89326878 under selection in PNG lowlanders.** This variant is in high-LD ($r^2 = 0.94$) with an introgressed ambiguous haplotype.

## Discussion

Our analysis of selective pressures in Papuan highlanders suggests that top-selected regions encompass genes that might have contributed to counteracting the detrimental effect of hypoxia in PNG highlanders and that candidate selection SNPs show associations with blood-related phenotypes. For example, the genomic region on chr12 overlapping with the gene *NCAPD2* demonstrates how hypoxic pressure may have impacted the genome and phenotypes of PNG highlanders. This region shows the third-highest XP-EHH score in PNG highlanders (Table 1, Fig. 1). The candidate SNP for this region, rs74576183-A>G (Fig. 3a), overlaps with the gene *NCAPD2* that is involved in various neurodevelopmental disorders[40,41]. Similarly, genomic regions under selection in Andeans living at intermediate altitudes show enrichment for neuronal-related genes, which might protect their brain from hypoxic damage[16]. Indeed, hypoxia at altitude impacts brain development and function when exposed during perinatal life[42] or long after birth[43]. This derived allele of the candidate SNP shows a significant association with increasing red blood cell count in the UK Biobank (Supplementary Table 10) and a suggestive association with slower heart rate from phenotypes measured in PNG (Fig. 3b, Supplementary

Table 13). Both these association results support adaptation through some cardiovascular-related processes in PNG highlanders, as we have already suggested when exploring the phenotypic differences between PNG highlanders and lowlanders[17]. The fact that this SNP shows significant eQTL associations and overlaps with open chromatin in multiple tissues would support its role in gene expression regulation. However, because this SNPs is in high LD with a missense variant with high DAF in PNG highlanders but rare in other populations (Supplementary Table 18), it is also possible that the real target for selection might be the missense variant (TAPBPL-G151R) that leads to changes in the TAPBPL protein and that is associated with antigen processing.

Similarly, the region under selection in PNG lowlanders containing the gene *SEC31A* and rs4693058-C>T, the candidate SNP for this region (Fig. 4a), is of particular interest to selection for pathogenic pressure in PNG lowlanders. Indeed, *SEC31A*[44] might play a role in immune processes, and the derived allele under selection of rs4693058-C>T, the candidate SNP for this locus, shows a significant association with various white cell percentage and count traits (Supplementary Table 11). Interestingly, the derived allele T under selection of rs4693058-C>T

shows a suggestive association with faster heart rate (Fig. 4b). But once again, we suggest that heart rate might be a proxy for other phenotypes (here, the white cells count[45]). Because rs4693058-C>T shows significant eQTLs and overlaps with open chromatin states in multiple tissues (Supplementary Table 15, Supplementary Fig. 14), we hypothesize that it impacts gene expression regulation.

The region with the highest XP-EHH, PBS and Fisher Score in PNG lowlanders (Fig. 2, Table 2, Supplementary Table 5) includes several genes from the guanine-binding protein (GBP) associated with immunity to diverse pathogens[34]. The association between a GBP7 variant and higher malaria symptoms has been reported in the Cameroon population[46], suggesting that this region might be selected due to malaria. The candidate SNP, rs368120563-T>C, is in LD with a missense variant (GBP2-A549T) and shows a high DAF in PNG lowlanders (DAF = 0.82) but is absent in non-Papuan populations (Supplementary Table 19). This particular missense variant might be the causal SNP, and selection might have targeted a change in the GBP2 protein sequence. This GBP locus carries an introgressed Denisovan-like haplotype with a frequency of 0.87 in PNG lowlanders. This introgressed haplotype includes both the candidate variant of the region (rs368120563-T>C) (Supplementary Table 21, Fig. 5) and the missense variant (GBP2-A549T) in PNG populations (Supplementary Fig. 15). Moreover, the missense variant can be found in the Denisovan genome (Supplementary Table 19, Supplementary Fig. 15) but the candidate SNP is not present in the Denisovan or any of the high coverage Neanderthal genomes. That pattern is compatible with the scenario where the candidate variant appeared after the introgression and where the introgressed haplotype frequency increased in the PNG populations driven by the selection acting on this variant. The alternative hypothesis would be that the candidate variant is not the target of selection (most likely the missense variant is), and the candidate variant is hitchhiked with the selected and introgressed haplotype.

We failed to find any candidate SNP associated with blood disorders (e.g., thalassemia) that are observed in high frequency in PNG lowlanders[47].

In this paper, we investigated selection in PNG highlanders and PNG lowlanders within 21 and 23 genomic regions under positive selection, respectively. We identified the SNP that most likely drives selection within each candidate region under selection and explored their association with several phenotypes measured within our dataset or the UK Biobank summary statistics. In both populations, several top SNPs were also significantly associated with several blood composition phenotypes in the UK Biobank. Within those significantly associated SNPs, one SNP driving selection in highlanders was associated with red blood cell count, and one SNP under selection in lowlanders was associated with lymphocyte percentage, suggesting associations with heart rate. Interestingly, when we examined the highest significant association of each candidate SNP with blood count phenotypes in the UKBiobank (Supplementary Tables 10–12), we observed an intriguing pattern in the distribution of the blood composition traits. Specifically, red blood cell traits stand out in their association with the candidate SNP of PNG highlanders, supporting that hypoxia might indeed been one of the main driving forces of selection that has acted on PNG highlanders. In contrast, PNG lowlanders candidate SNPs are either associated with white blood cells or platelets, hinting at specific environmental pathogenic pressures (e.g., malaria[48]) that might have shaped the genome of PNG lowlanders. Further studies will be needed to clarify the complexity of the haematological responses to hypoxia and pathogenic pressures in PNG, as many of these SNPs can affect multiple phenotypes and which phenotype might be the true driving force is beyond the scope of this study. We found that three candidate SNPs for PNG highlanders and one for PNG lowlanders are in high LD with the introgressed haplotypes suggesting adaptive introgression. Our results suggest that selection in PNG highlanders and lowlanders

has partially targeted introgressed haplotypes from Neanderthals and Denisovans. This study demonstrates that both PNG highlanders and PNG lowlanders carry signatures of positive selection and that the associated phenotypes largely match the challenges they faced due to their respective environments.

## Methods

### Ethics
This study was approved by the Medical Research Advisory Committee of Papua New Guinea under research ethics clearance MRAC 16.21 and the French Ethics Committees (Committees of Protection of Persons CPP 25/21_3, n_SI: 21.01.21.42754). Permission to conduct research in PNG was granted by the National Research Institute (visa n°99902292358) with full support from the School of Humanities and Social Sciences, University of Papua New Guinea. All samples were collected from healthy, unrelated adult donors who provided written informed consent. After a full presentation of the project to a wide audience, a discussion with each individual willing to participate ensured that the project was fully understood. We did not provide any compensation to the participants. No research findings from our study apply to only one sex or gender. The participants included males and females, and the cohort did not include or exclude participants based on gender. In our study, gender was not considered in the study design. The gender of participants was determined based on self-report.

### Samples
DNA was extracted from saliva samples with the Oragene sampling kit (DNA Genotek Inc., Ottawa, Canada) according to the manufacturer's instructions. Sequencing libraries were prepared using the TruSeq DNA PCR-Free HT kit (Illumina, Inc., San Diego, California, USA). About 150-bp paired-end sequencing was performed on the Illumina HiSeq X5 sequencer (Illumina, Inc., San Diego, California, USA). We sequenced PNG whole genomes from PNG lowlanders from Daru Island (n = 80, <100 m above sea level (a.s.l)), PNG highlanders from Mount Wilhelm villages (n = 60, 2300 and 2700 m a.s.l.) and individuals sampled in Port Moresby from different origins (n = 64) - that we designated as PNG diversity set I - sampled between 2016 and 2019 (EGAD00001010142, EGAD00001010143, EGAD50000000050). We also included 58 already published PNG genomes[2] (EGAS00001005393) in the PNG diversity set I, increasing its sampling size to 122 individuals (Supplementary Note 1, Supplementary Data 1 and 2). We measured phenotypes associated with body proportion, pulmonary capacities and cardiovascular components in these three PNG datasets (PNG highlanders, PNG lowlanders, PNG diversity set I)[17] (Supplementary Note 2, Supplementary Table 2).

We combined these 262 sequences with published Papuan genomes (n = 81, PNG diversity II)[26,39,49–51] and high-coverage genomes from the 1000 Genomes project from Africa (n = 207), East Asia (n = 202) and Europe (n = 190)[52]. To better describe the genetic structure of the studied populations from PNG, we also added the Australian genomes of the SGDP project (n = 2)[51] and East and West Island Southeast Asia (ISEA) (n = 71)[26].

### Variant calling
Sequencing data for all samples used in this study were processed together, starting from the raw reads. FASTQ files were trimmed with fastp v0.23.2[53] and converted to BAM using Picard Tools FastqToSam v2.26.2[54]. Further processing was performed with Broad Institute's GATK Germline short variant discovery (SNPs and Indels) Best Practices[55]. HaplotypeCaller tool was used to produce individual sample GVCF files, which were further combined by JointGenotyping workflow to create multi-sample VCF files. GATK v4.2.0.0 was used[56]. Data were processed with GRCh38 genome reference (Supplementary Note 3).

## Filtering

Unless otherwise stated, we performed the analysis on biallelic SNPs with a maximal missing rate of 5% that remained after genomic masking (Supplementary Note 6). For each pair of related individuals to the second degree, when relevant, we kept the individuals with the highest number of phenotype measurements or the individual with the highest mean of coverage. We removed two PNG samples with low call rates from any further analysis. Quality and kinship filtering resulted in 249 unrelated genomes among the PNG highlanders, lowlanders and the PNG diversity set I: 54 sequences from PNG highlanders, 74 sequences from PNG lowlanders and 121 sequences from individuals originating from different parts of PNG and sampled in Port Moresby (PNG diversity set I; Supplementary Notes 1, 4-5, Supplementary Table 1, Supplementary Data 1–3, Supplementary Figs. 1 and 2). The unrelated and filtered dataset also includes published sequences from New Guinea ($n = 81$, PNG diversity II)[26,39,49–51], Africa ($n = 207$)[52], East Asia ($n = 202$)[52], Europe ($n = 190$)[52], Australia ($n = 2$)[51], East and West Island Southeast Asia ($n = 71$)[26] (Supplementary Note 1; Supplementary Data 1).

## Population structure

Principal Component Analysis (PCA) was performed on the unrelated dataset filter for variants with minor allele frequency lower than 0.05 and pruned for linkage disequilibrium using the smartpca program from the EIGENSOFT v.7.2.0 package[57]. The LD pruned dataset included 456,379 SNPs (4,751,609 SNPs before pruning). We computed the PCA to the third principal component. We ran ADMIXTURE v1.3[58] on the same dataset from components K = 2 to K = 10 (Supplementary Note 7). For each component, ADMIXTURE computes the cross-validation error using a k-fold cross-validation procedure. We set the k parameter to 100. In order to assess the fit of each model generated, we generated the cross-validation error ten times for each component. We then defined the confidence interval of the cross-validation error for the component using the quantiles of the ten generated cross-validation errors.

In order to explore further the extent of admixture in PNG low-landers, we performed three D statistic tests using the qpDstat command from admixtools v7.0.1[59]. We computed D statistic tests of the form D(W,X,Y,Z) for either PNG highlanders or PNG diversity set I for population W and PNG lowlanders or PNG diversity set I for population X. In each case, we used PNG highlanders, CHB, and YRI populations for populations W, Y, and Z.

## Phasing

We phased genomes from Mt Wilhelm, Daru, PNG diversity set I, Africa, Asia and Europe using shapeitv4.2.2[60]. We phased the samples statistically without reference, as the reference haplotypes panel for the PNG population does not exist (Supplementary Note 8).

## Selection analysis

We aimed to identify genomic regions carrying signatures of positive selection in PNG highlanders and lowlanders using three complementary approaches. We computed the Population Branch Statistic (PBS), a method based on allele frequency differentiation, to detect natural selection signals in PNG highlanders and lowlanders[61]. For the PBS scores in PNG highlanders, we used PNG lowlanders as the reference population and YRI from the 1000 Genomes Project as the outgroup. When performing PBS on PNG lowlanders, we used PNG highlanders as the reference population and the YRI as the outgroup. We chose YRI as the outgroup because we were interested in exploring potential introgressed haplotypes within the candidate regions for selection, and we wanted to avoid masking adaptative introgressed regions common between PNG and non-African populations. In both cases, we obtained a PBS score for every biallelic SNP. We then defined sliding windows of 20 SNPs with a step of 5 SNPs to identify multiple

adjacent SNPs with an elevated PBS score (which lowers the random chances due to drift). We assigned the average PBS score of all the SNPs included in the sliding window as the PBS score of the window. We kept the sliding windows with an average PBS score in the 99th percentile and merged the top sliding windows that are 10 kb maximum from each other. The top PBS score of the sliding windows in the region was given to the whole merged region.

In order to detect more recent selection, we computed the cross-extended haplotype homozygosity (XP-EHH) on the phased dataset with selscan v2.0.0[62] to test for positive selection using haplotype information. We computed XP-EHH for every SNP using PNG highlanders as the target population and PNG lowlanders as the reference population. While the maximal scores define SNPs under selection in PNG highlanders, the lowest scores indicate the SNPs under selection in PNG lowlanders. We determined the top SNPs for XP-EHH score in PNG highlanders as the SNP with XP-EHH score in the 99th percentile. We kept the SNPs whose XP-EHH score was in the 1st percentile for PNG lowlanders. We merged these top SNPs in windows so that two top SNPs distant by at most 10 kb are included in the same window. This merging step results in windows whose endpoints are the two most distant top SNPs included in the window.

Next, we combined the PBS and XP-EHH scores in a Fisher score[25]. We used the sliding windows of 20 SNPs and 5 SNPs step defined for the PBS score. For each of these sliding windows, we gave as XP-EHH score the highest XP-EHH score among the 20 SNPs included in the windows. We combined the PBS and XP-EHH scores in a Fisher Score ($-\log_{10}(PBS_{percentilrank}) - \log_{10}(XP - EHH_{percentilrank})$) for each sliding window. Finally, we selected the windows with Fisher Score in the 99th percentile and merged them when they were distant by a maximum of 10 kb.

We used a random sampling approach to test the significance of the top 10 windows with the highest score for each of the three selection scores. We looked if the score of the unit used to determine the score of the genomic region – SNP for XP-EHH or 20-SNP windows for PBS and Fisher score – with the highest score in the region was significantly higher than the score of random units along the genome (Supplementary Note 9).

We extended the top 10 windows with the highest score for each of the three methods by a 50 kb flanking region. Finally, we merged the regions from these 30 top regions that overlapped to obtain the final non-overlapping regions of interest that we will use further.

Because of the low number of individuals per population, the high genetic diversity in PNG, and the substantial contribution from East Asian and ISEA ancestry in the PNG diversity sets I and II (Supplementary Figs. 3 and 4), we did not include these samples in the selection analyses described above.

Due to the uneven distribution of mean depth of coverage in our sample interval (Supplementary Data 2, Supplementary Fig. 1), we have checked for the impact of including individuals with higher mean coverage in the selection scans. We performed the XP-EHH and PBS selection scans on PNG highlanders and lowlanders while we have reduced the coverage for the 15 PNG lowlanders outliers with a mean of coverage higher than 25 to 30% of their previous coverage (Supplementary Note 10).

## Selection of the candidate SNPs

We computed ancestral recombination graphs for the phased dataset with RELATE v1.1.8[63] (Supplementary Note 12, Supplementary Fig. 12). We generated coalescence rates through time within PNG highlanders and lowlanders from their respective subtrees. Finally, we extracted the local tree for each SNP in the regions of interest from PNG highlanders and lowlander subtrees. We used these local trees as input for Coalescent Likelihood Under Effects of Selection (CLUES) (v1)[35] (Supplementary Note 13). CLUES assigns a likelihood ratio (logLR) to each SNP of interest that reflects the support for the non-neutral model. For

each SNP in the region of interest, we computed logLR five times by resampling the local tree branch length and averaged the logLR for the five runs. To decide between the top five SNPs with the higher average logLR in each genomic region, we generated the logLR 50 additional times for these five SNPs. We considered the SNP with the highest average logLR after 50 runs as the SNP the most likely to drive selection within the regions under selection (i.e., candidate SNPs). Because SNPs with low DAF (Derived Allele Frequency) are unlikely to be under selection, we did not consider SNPs with DAF lower than 0.05. We also filtered out fixed variants for which CLUES cannot compute the logLR.

### Association in the UK Biobank
To further understand how the candidate SNPs affect phenotypes, we downloaded the UK Biobank's summary statistics[64] for the 1931 phenotypes with more than 10,000 samples. We excluded phenotypes associated with sociocultural influences and local environmental factors due to the lack of correspondence between the environmental variables grouped under these phenotypic categories and the actual environmental conditions experienced by the studied Papua New Guinean populations. (Supplementary Note 14). We extracted the p-value and the beta of the candidate SNPs for each of the 1470 remaining phenotypes. To avoid the ancestry sample size bias present in UK Biobank, we only extracted the p-value (pval_EUR) and beta score (beta_EUR) for European ancestry. Because the PNG population has a unique genetic diversity that is absent in Europeans, some candidate SNPs were not listed in the UK Biobank. In that case, we looked for summary statistics for the closet SNP from a 50 bp upstream and 50 bp downstream region. After extracting the SNP summary statistics for every phenotype, we only consider the phenotype of interest if the $\log_{10}(p-value)$ is lower than −10.47 to correct for multiple testing considering the significance threshold of $\log_{10}(5 X 10^{-8})$ that needs to be corrected for the number of phenotypes studied ($\log_{10}\frac{5X10^{-8}}{1470}$).

We generated a null distribution by randomly sampling $x$ windows ($x$ being the number of candidate SNPs with at least one significant association in PNG highlanders or lowlanders) among 100 bp windows – following the 50 bp upstream and downstream closest SNP approach – associated with at least one phenotype in the UK Biobank. We then checked how many of the $x$ random windows include at least one SNP associated with at least one blood phenotype (from the "Blood count" category 100081 of the UK Biobank). We resampled 10,000 sets of x windows. To test for significance, we computed a resampling p-value by calculating the proportion of random window sets in which the number of associations with at least one blood phenotype was higher than that observed windows associated with one blood phenotype.

### Association test
We used Genome-wide Efficient Mixed Model Association (GEMMA) (v0.98.4)[65] to detect if the candidate SNPs are associated with any phenotypes we measured in the PNG highlanders, lowlanders and PNG diversity set I datasets (Supplementary Note 15). We corrected the haemoglobin concentration, blood pressure, heart rate and BMI for age and gender and the chest depth, waist circumference, weight, and pulmonary function measurements (FEV1, PEF and FVC) for age, gender and height using a multiple linear regression approach[17].

We performed association tests with the GEMMA univariate Linear Mixed Model (LMM) for the candidate SNPs and each corrected phenotype. To increase our sampling size, we performed these association tests using all the PNG individuals (highlanders, lowlanders and PNG diversity set I) with at least one phenotype measurement ($n = 234$) (Supplementary Table 2). We incorporated the centred relatedness matrix computed with GEMMA into the LMM, using all the 234 PNG sequences to correct for population stratification. We corrected each p-value for the number of SNPs tested with the Benjamini-Hochberg procedure. Because these phenotypes can be gathered in five groups

of highly correlated phenotypes[17], we used a threshold for significance of 0.01 (0.05/5) to correct for the number of phenotypes tested.

### Introgression
We scanned all selection candidate regions for introgressed archaic fragments using a previously published Hidden-Markov-Model (hmmix) method[38]. We ran hmmix on each PNG highlanders and lowlanders sample using default parameters and the 1000 Genomes Project YRI population as the unadmixed outgroup. To define introgressed haplotypes in each PNG highlander and lowlander individual, we kept all segments annotated as "Archaic" by hmmix, with an average posterior probability (mean_prob) larger than 0.8, a recommended threshold by the authors of this method. In addition, we required the segment to fully overlap with the candidate regions and share at least one archaic SNP (aSNP) with any of the four archaics (Altai[66], Vindija[67], and Chagyrskaya[68] Neanderthals and the Denisovan[23]). An aSNP is defined as a SNP absent in the Yoruba population that falls within the borders of an inferred haplotype. All four archaic genomes were filtered using suggested genomic masks stored with raw genomes. We then compared introgressed haplotypes to different archaic ancestries based on their sequence similarity to the genomes of three Neanderthals and the Denisovan. Here, we relied on the detected archaic SNPs by the HMM and their presence in the genomes of the four archaic individuals. More specifically, if a haplotype shared more aSNPs with Denisovan than with all Neanderthals, we referred to it as Denisova. Inversely, we annotated haplotypes with more shared aSNPs with all Neanderthals than the Denisovan as Neanderthal. All other haplotypes were annotated as ambiguous.

In order to count haplotypes' frequencies across PNG populations, we combined results from the individual-level data into regions of introgression using highly linked aSNPs ($r^2 > 0.5$) shared across individuals' haplotypes. Per each candidate region for selection, we reported the introgressed haplotype with the highest frequency in the corresponding PNG population where the genomic region is found under selection. In instances where we found evidence for individuals' haplotypes of different archaic ancestries in a given introgressed region, we reported the most frequent ancestry.

We generated a median-joining haplotype network for 10 kb windows centred on the candidate SNP when we found that the candidate SNP was in high-LD ($r^2 > 0.5$) with at least one of the archaic SNPs of the introgressed haplotype (Supplementary Note 16).

### Prediction of variant effect
As an additional effort to decipher the function of the candidate SNPs (e.g. gene expression or changes in protein sequence), we looked for significant eQTLs for each candidate SNP using the Genotype-Tissue Expression (GTEx) Portal[69]. In addition, we downloaded the 111 reference human epigenomes from the Roadmap Epigenomics project[70] to explore which chromatin state the candidate SNPs fall in different tissue types. Finally, we used The Ensembl Variant Effect Predictor (VEP)[71] on the region under selection to detect missense variants in these regions with the canonical flag.

Unless stated otherwise, we generated the main and Supplementary Figures with matplotlib v3.5.3 and seaborn v0.12.2 packages under python v3.7.12

### Reporting summary
Further information on research design is available in the Nature Portfolio Reporting Summary linked to this article.

## Data availability
The whole genome sequences and the phenotype measurements from the 60 PNG highlanders, 80 PNG lowlanders, and 64 individuals sampled in Port Moresby generated in this study have been deposited in the European Genome-phenome Archive under accession codes

EGAD00001010142, EGAD00001010143 and EGAD50000000050. The whole genome sequences are available under restricted access to protect the privacy of the participants, in agreement with the Institutional Review Board approval and the individuals' informed consent forms. The data are available to the scientific community under controlled access and reviewed by the Data Access Committee of the Papua New Guinean Genome Diversity Project. Access to the new whole genome sequences published with this paper is automatically granted upon request in the case of replication of the published study. New demographic, selection and association studies require approval from the Papua New Guinean Genome Project program (PNGP) committee. Source data generated in this paper can be accessed on the linked figshare repository [https://doi.org/10.6084/m9.figshare.23695062][72].

Published samples used in this study were retrieved from the European Nucleotide Archive (ENA) under the accession numbers PRJEB9586 and ERP010710[51] and PRJEB6463[49]; the European Genome-phenome Archive (EGA) under the accession numbers EGAS00001001247[50], EGAS00001003054[26] and EGAS00001005393][2]; the database of Genotypes and Phenotypes (dbGaP) under the accession number phs001085.v1.p1[39]. The high-coverage genomes from the 1000 Genomes project were accessed at https://www.internationalgenome.org/data-portal/data-collection/30x-grch38[52].

The GRCh38 genome reference was built into the GATK Germline short variant discovery (SNPs + Indels) workflow. The resources required for this workflow are accessible on the following Google Cloud bucket [https://console.cloud.google.com/storage/browser/genomics-public-data/resources/broad/hg38/v0/][55].

The Ancestral Genome for Homo sapiens (GRCh38) was retrieved from the Ensembl website [https://ftp.ensembl.org/pub/release-93/fasta/ancestral_alleles/homo_sapiens_ancestor_GRCh38/][71]. The genetic map for GRCh38 was accessed from Eagle website [https://alkesgroup.broadinstitute.org/Eagle/downloads/tables/]. UK biobank GWAS summary statistics are available at http://www.nealelab.is/uk-biobank[64].

## Code availability
All custom codes used in this study are available on GitHub (https://github.com/mathilde999/selection-png)[73].

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

## Acknowledgements

Whole genome sequences from Daru, Mt. Wilhelm and Port Moresby were generated at the National Centre of Human Genomics Research (France) or the KCCG Sequencing Laboratory (Garvan Institute of Medical Research, Australia). The authors thank Ray Tobler (Australian National University), Roxanne Tsang (Centre for Social and Cultural Research, Griffith University, Australia), Kylie Sesuki and Teppsy Beni (School of Humanities and Social Sciences, University of Papua New Guinea), and Alois Kuaso and Kenneth Miamba (National Museum and Art Gallery, Papua New Guinea) for their help during the sampling campaigns. We especially thank all of our study participants. Data analyses were carried out in part in the High-Performance Computing Centre of the University of Tartu. This research was supported by the European Union through the European Regional Development Fund (Project No. 2014-2020.4.01.16-0030) to M.A. and F.M.; and through Horizon 2020 research and innovation programme under grant no 810645 and the European Regional Development Fund project no. MOBEC008 to M.A., G.H., V.P., D.Y., R.K., M.D., T.O., M.Metspalu and M.Mondal; the French Ministry of Foreign and European Affairs (https://www.diplomatie.gouv.fr) (French Prehistoric Mission in Papua New Guinea) and the French Ministry of Research grant Agence Nationale de la Recherche (https://anr.fr) number ANR-20-CE12-0003-01 to F.X.R.; the LabEx TULIP, France (https://www.labex-tulip.fr) to F.X.R. and N.B.; the Leakey Foundation (https://leakeyfoundation.org) to N.B. and the Fondazione con il Sud (2018-PDR-01136) to F.M. This study was also supported by the French Embassy in Papua New Guinea

(https://pg.ambafrance.org), and the University of Papua New Guinea, Archaeology Laboratory Group. The CNRGH sequencing platform was supported by the "France Génomique" national infrastructure, funded as part of the "Investissements d'Avenir" program managed by the "Agence Nationale pour la Recherche" (contract ANR-10-INBS-09).

## Author contributions
F.-X.R., N.B., M.L., T.O. and M.Metspalu designed the study. F.-X.R., N.B., M.L., J.K., N.E. and J.M. collected the data. V.M., A.B., and J.F.D. generated whole-genome sequences. M.A., N.B., G.H., V.P., D.Y., F.M., R.K., M.D. and M.Mondal performed the data analysis. F.-X.R., N.B., M.Metspalu and M.P.C. provided resources and logistics. M.A., N.B., M.Mondal. and F-X.R. wrote the manuscript with the contribution of all the co-authors.

## Competing interests
The authors declare no competing interests.

## Ethics approval
Local researchers were involved throughout every step of the research process: study design, study implementation, data ownership, intellectual property and authorship of publications. This study has been determined in collaboration with local partners and approved by a local ethics review committee.
