## [Peer Review File · Nature Communications]

Positive selection in the genomes of two Papua New Guinean populations at distinct altitude levelsREVIEWER COMMENTS

Reviewer #1 (Remarks to the Author):

This paper describes a selection scan in the genomes of people from Papua New Guinea. Ultimately, they identify some signals and suggest that the high/low altitude groups experienced selection for cardiovascular/immune phenotypes respectively. I think it's an interesting hypothesis that there might be some sort of tradeoff between living at high altitude versus in areas with high disease burden.

On the other hand, I felt the claims weren't really fully justified. There is no formal test of significance for individual loci so we can't really fully evaluate how strong the evidence is for the signals. There's also no formal test for enrichment of particular types of signals, so we can't really tell whether the claimed enrichments in cardiovascular-related and immune-genes at high and low latitude respectively are significant. In general, the top few signals that are shared across sweeps (GBP2, LINC02388) look more convincing and the paper could perhaps justify focusing on those more.

Overall, I find the results suggestive but not really fully supported by this analysis.

Major comments.

1) The definition of the candidate selected regions (top 10 for each statistic) is a bit arbitrary. Why 10? The problem with looking at outliers is that we don't know whether they are actually significantly different (even if there were no selection at all you would identify the same number) and there are no tests of significance. I find the top few signals that are shared between different statistics to be more convincing since the statistics are somewhat (but not entirely) independent. I get that it's difficult to construct fully convincing tests of significance, but one can at least try to construct something that's calibrated genome-wide.

2) There's no formal test of enrichment so it's hard to know whether the selection for haematological phenotypes in the high-altitude population is actually significant or just anecdotal. UK biobank is extremely well-powered for these phenotypes, so there will be a lot of associations in any set of regions. I suppose the appropriate test would be to compare the number of associations in matched sets of regions (matched on gene count, conservation etc...), or those identified as under selection in other populations.

3) Similarly, there's no formal test of whether there is an enrichment in archaic introgression or whether it's about what one would expect given the overall high levels of archaic introgression in these populations.

4) The authors do mention that associations in UK biobank might not transfer to the PNG populations but I think this is actually a pretty serious issue. How can we interpret these associations? The authors could look to see whether these specific associations replicate nominally in East Asian and African ancestry individuals from UKB.

5) is 1600-2400 M actually high enough to cause a substantial selective pressure related to altitude? The cited reference 8 says the following which suggest perhaps not:

“Although the expression ‘high altitude’ has no precise definition, the majority of individuals have certain clinical, physiological, anatomical and biochemical changes which occur at levels above 3000 m [~9,900 ft]. Individual variation is, however, considerable, and some people are affected at levels as low as 2000 m [~6,600 ft]” (West et al. 2007, p. 27; notes in brackets are added for clarification). Scientists often choose altitudes of 2,500 m (~8,250 ft) and above as a working definition of high altitude. ”

Most of the papers cited in the first paragraph of the introduction refer to populations at much greater altitudes (e.g. Tibetans, Sherpas, Andeans).

6) Malaria seems like a much more plausible selective pressure but on the other hand it is already well-established for several decades that it has caused differential selection in these populations for example at globin genes – see Flint et al (1986) [<https://www.nature.com/articles/321744a0>] for example specifically in PNG. I'm surprised that this and related papers are not cited. Indeed, I wonder why the authors do not see signals of selection at Malaria related genes since as they themselves say, that is likely to be one of if not the strongest disease selection pressure.

7) It is really hard to read the haplostrips figures in the main text supplement and see what the archaic haplotypes are. Perhaps a haplotype network would be easier to read and complement these figures.

8) Why use YRI as an outgroup for the PBS instead of say, Europeans who should also be an outgroup but would share the OOA bottleneck and be much closer in terms of genetic distance.

Typos etc

Line 420: "Archaic introgressions" should be "Archaic introgression"?

Line 451 "Cardio Vascular" should be "Cardiovascular phenotypes" or something?

Reviewer #2 (Remarks to the Author):

This study aims at detecting selection signals in two genetically closely related populations: highlanders and lowlanders from Papua New Guinea (PNG). The authors hypothesize that distinct environmental constraints have induced specific selective pressures on the two populations, which lead to unique genetic changes and phenotypic differences. The authors performed whole genome sequencing on 54 PNG highlanders and 74 PNG lowlanders. They performed a genome-wide selection scan and found several candidate genomic regions and SNPs under selection. The authors claim that many of these candidate SNPs play a role in gene expression or gene function and show association with cardiovascular-related and pathogenic-defense-related phenotypes in PNG highlanders and lowlanders, respectively. They also claim that many of the candidate genomic regions under selection harbor archaic introgression.

I think detecting selection signals in PNG highlanders and lowlanders is generally a very interesting and important topic. However, I have several major concerns about the analysis and conclusions reported in the study that need to be addressed before publication.

1. Methods used for selection scan are invalid and could lead to false positives.

To infer the population structure of PNG individuals, the authors perform PCA and ADMIXTURE analysis using the merged dataset including newly sequenced PNG individuals, published Papuan genomes, and high-coverage genomes from the 1000 Genomes project from Africa ($n=207$), East Asia ($n=202$) and Europe ($n=190$). While this shows, PNG samples are very distinct from 1000 Genomes populations, they do not help much to characterize the genetic structure in PNG or their relationship to other groups. To better characterize the genetic ancestry of PNG individuals, samples from other populations like Oceanians from Human Genome Diversity Panel should be added.

Both PCA and ADMIXTURE, however, show that PNG lowlanders in their dataset fall intermediate of PNG highlanders and PNG diversity panel and have ancestry from multiple groups when setting $K > 3$, suggesting admixture. However, the authors do not explore the demographic history of PNG highlanders and lowlanders. Admixture would have a major impact on the selection scan. Moreover, the PNG population is highly drifted (Brucato et al. 2021) which could also impact the power and false discovery rate of selection scans. The authors should perform formal tests of admixture, e.g., f -statistics, treeMix, etc. to explore the history of admixture and founder events in PNG groups.

Importantly, if PNG lowlanders are admixed (as ADMIXTURE plots suggest), both PBS and XP-EHH could lead to false positives as the methods assume the test populations are homogenous. For example, the deviation of allele frequency detected in PBS by comparing PNG highlanders and PNG lowlanders could simply highlight regions of unknown ancestry in lowlands and not selection and the long haplotypes detected by XP-EHH may simply be due to signals of recent admixture. The authors need to first characterize the demographic history of the two populations and conduct selection scans with methods that account for admixture, like the method from Sanchez et al. (2013) (<https://doi.org/10.1093/molbev/mst089>).

2. Sanity checks for selection scans needed.

Data QC: It is important that the authors include some quality metrics to show the reliability of their data (e.g., sample coverage, genotyping, and mapping quality, transition-transversion ratios, rates of heterozygosity, etc.). See 1000 Genomes and other genome coverage papers for commonly reported metrics. In addition, the authors should infer the phasing error rates and provide some evaluation of how phasing error, genotyping error, and variable coverage across samples impact their results. As shown in Figure S2, the samples used in this study have an uneven distribution of coverage: PNG highlander samples have higher mean coverage than PNG lowlander samples. The authors should show that the selection signals are not biased by the uneven sample coverage across the two populations and that the selection signals are not limited to specific sets of samples of high/low coverage.

Analysis QC: To detect genomic regions under selection, the authors first perform PBS on PNG highlanders and lowlanders setting YRI as an outgroup. They then perform XP-EHH and finally combine PBS and XP-EHH scores with Fisher score. For all these three steps, the authors keep regions with scores fall in 99th percentile and take the intercept of the three sets as the final candidate genomic regions. They then perform CLUES to find candidate SNPs driving selection in these regions. I find the pipeline for selection scan to be very puzzling: although the authors have combined PBS and XP-EHH results with Fisher score, they seem to still use 30 hits that are the top 10 in each criteria. Shouldn't they consider only the top hits based on Fisher score?

Moreover, the authors use Relate for inference of the ARGs. Recent studies have shown that Relate and other tree-based methods can have large uncertainty and errors in the inference of mutation ages (Brandt et al. 2022). Do the authors account this uncertainty in the CLUES analysis? How do these estimates impact their results? It would be useful if the authors can provide simulations to show that CLUES provides reliable results under their setup, especially for populations matching the demographic history of Papuans. Moreover, the authors need to at least include some sanity checks on their results: as positive controls, they should show if CLUES was able to detect well-known selection signals like LCT in EURs in their dataset and provide reasonable selection coefficient estimation. The authors should clearly state all parameters they input to CLUES. They should also provide results of Relate to show if the coalescent rates of the Papuan populations look reasonable.

3. Rigorous association tests needed.

The authors use UK biobank summary statistics for EURs to look for significant phenotype association with candidate SNPs driving selection in their dataset. I think it is not a good idea to use genome-wide associations from UK biobank European ancestry since the authors have shown that PNGs have no European ancestry component in ADMIXTURE results and many studies have shown that GWAS results are not transferable across populations (as noted by the authors themselves). The authors instead use or at least show robustness to the use of genome-wide association statistics inferred from other populations such as Biobank Japan.

The authors also apply GEMMA to study the association between candidate SNPs and the phenotypes they measured, correcting for age, sex and height. Importantly, however, they do not correct for population structure even as they combine very diverse PNG lowlands and highland samples. Moreover, as mentioned in lines 383-385, they did not find any significant phenotype

association for top selection candidate SNPs when correcting for the number of SNPs and phenotypes tested together. This makes the result discussed in lines 361-379 puzzling and questionable since all of them are based on direct association tests of each SNP with the phenotype. The authors argue that this may be because of the low sample size, which might well be true but even without correction there are no significant hits and so this seems very speculative.

4. Inference of archaic ancestry.

The authors apply haplostrips to scan for regions with archaic haplotypes in candidate genomic regions under selection. This is not a rigorous way of looking for archaic ancestry in human genomes. To my understanding, haplostrips is a visualization method, no statistical tests are conducted on top of the clustering to show the actual probability of a genomic region coming from a specific ancestry. The authors should apply methods such as IBDmix (Chen et al. 2020), Sprime (Browning et al. 2018), HMM from Skov et al. (2018), DICAL-ADMIX (Steinrucken et al. 2018), etc. to infer archaic ancestry from the PNG genomes and conduct statistical tests to show that the regions under selection are enriched for haplotypes from archaic populations in populations of PNG individuals. As a sanity check, it would be useful if the authors can also report the % of Neanderthal and Denisovan ancestry per individual in their study and use this information to assess if locally at their candidate loci there is significantly higher archaic ancestry.

5. Quantitative assessment of signals is lacking.

There are many paragraphs in the manuscript where the authors discuss evidence based on raw count numbers without any statistical test of enrichment or significance. Some examples include line 393-402, 403-419. It is important to either provide clear tests of significance or remove these arguments.

6. Release of data and scripts to ensure reproducibility.

Line 540-542 states that the authors would share their newly sequenced genomes upon publication. Would all the raw data and VCF files be deposited in EGA or other publicly accessible datasets? To ensure reproducibility of the main results, it is also important if the authors release all the data used for the analysis including the Papuan Diversity genomes (I tried to access this but this dataset is not available). Moreover, the authors should share their scripts for data processing and the selection scans (PBS, XP-EHH, CLUES, Relate, GEMMA, etc.) or clearly state all parameters and source files that were used. For example, it is unclear what human ancestor sequence did the authors use to polarize ancestral / derived alleles. And what parameters or filters were applied for various analysis.

Minor points:

1. The authors found 21 regions after selection scan. However, they don't keep the same set of candidate regions in all following analyses. For example, in the section on looking for archaic segments they use 44 genomic regions instead of 21.
2. In Supplementary Note 1c, the authors use (The 1000 Genomes Project Consortium et al., 2015) as the reference for their 1000 Genome high coverage data. This is confusing since this paper only releases low coverage Phase III 1000 Genome data based on my understanding. They should probably cite (<https://doi.org/10.1016/j.cell.2022.08.004>) instead.
3. Typo "hypothesise" in the abstract.
4. Line 136 - sampled between 2016 and 2019 (EGA accession code XXXXX). Update EGA accession number.
5. Line 440-441: This fact and the gene flow between the Altai Neanderthal and Denisova would suggest that we most likely observed Denisovan introgression within the GBP locus in the PNG population. Is there evidence that Altai Neanderthal has Denisova ancestry in this region?

REVIEWER COMMENTS

II. Reviewer #1 (Remarks to the Author):

This paper describes a selection scan in the genomes of people from Papua New Guinea. Ultimately, they identify some signals and suggest that the high/low altitude groups experienced selection for cardiovascular/immune phenotypes respectively. I think it's an interesting hypothesis that there might be some sort of tradeoff between living at high altitude versus in areas with high disease burden.

On the other hand, I felt the claims weren't really fully justified. There is no formal test of significance for individual loci so we can't really fully evaluate how strong the evidence is for the signals. There's also no formal test for enrichment of particular types of signals, so we can't really tell whether the claimed enrichments in cardiovascular-related and immune-genes at high and low latitude respectively are significant. In general, the top few signals that are shared across sweeps (GBP2, LINC02388) look more convincing and the paper could perhaps justify focusing on those more.

Overall, I find the results suggestive but not really fully supported by this analysis.

We thank the reviewer for their insightful comments. Below are our detailed responses to these different issues that we also addressed in the revised manuscript.

Major comments.

1) The definition of the candidate selected regions (top 10 for each statistic) is a bit arbitrary. Why 10? The problem with looking at outliers is that we don't know whether they are actually significantly different (even if there were no selection at all you would identify the same number) and there are no tests of significance. I find the top few signals that are shared between different statistics to be more convincing since the statistics are somewhat (but not entirely) independent. I get that it's difficult to construct fully convincing tests of significance, but one can at least try to construct something that's calibrated genome-wide.

We agree with the reviewer's comment that we need to test the significance of the candidate regions for positive selection. We used a random sampling approach to test the significance of the top 10 windows with the highest score for each of the three selection scores, following an approach resembling (Skoglund et al., 2017) methodology, we looked if the score of the unit used to determine the score of the genomic region - SNP for XP-EHH or 20-SNP windows for PBS and Fisher score - with the highest score in the region was significantly higher than the score of random units along the genome. For each selection test, we defined random units along all the autosomes so that they are separated by at least 5Mb, and thus remain independent. For XP-EHH, we considered the distribution of the XP-EHH of individual random SNPs that we compared to the score of the top SNP in each candidate region. For the PBS and Fisher scores, the random distribution was built using random sliding windows of 20 SNPs defined in the approach we used to select the top 10 regions. We compared this distribution to the PBS or Fisher score of the sliding windows with the highest score in each candidate region.

Finally, we computed a Z score for each candidate region (\$Z(x)\$ ) using the mean (\$\mu\$ ) and standard deviation for the score of the random regions (\$\sigma\$ ) and the score of the candidate region (\$f(x)\$ ). \$Z(x) = (f(x) - \mu) / \sigma\$.

We found that for every score, in both PNG highlanders and lowlanders, the 10 top regions have a significantly higher score than random regions (\$|Z \text{ score}| > 3\$ ). Although there are probably more than 10 significant regions per score, we decided to limit our analysis to the 10 top regions for each score because of the number of analyses that we had to carry on these regions (e.g., Clues, which is computationally intensive) which are unfeasible right now.

We described this random sampling further and added the results to the revised manuscript (Revised manuscript: page 9, line 269 – page 13, line 399 – Supplementary Tables 7 and 8 - Supplementary Note 9).

Skoglund, P., Thompson, J. C., Prendergast, M. E., Mittnik, A., Sirak, K., Hajdinjak, M., ... Reich, D. (2017). Reconstructing Prehistoric African Population Structure. *Cell*, 171(1), 59-71.e21. <https://doi.org/10.1016/j.cell.2017.08.049>

2) There's no formal test of enrichment so it's hard to know whether the selection for haematological phenotypes in the high-altitude population is actually significant or just anecdotal. UK biobank is extremely well-powered for these phenotypes, so there will be a lot of associations in any set of regions. I suppose the appropriate test would be to compare the number of associations in matched sets of regions (matched on gene count, conservation etc...), or those identified as under selection in other populations.

In order to design a formal enrichment test for blood phenotype associations with our candidates, we employed a random sampling approach that compares candidate SNP associations to those of random background SNPs.

It is noteworthy that some candidate SNPs were not present in the UK biobank. In that case, we screened the biobank for summary statistics for the closest SNP from the candidate SNP. We restrained the limit to define the closest SNP to a candidate SNP upstream and downstream 100bp in case the SNP is not present in the UK biobank. We then observed that out of the 5 candidate SNPs that show significant association with at least one phenotype in the UKBB for PNG highlanders, 3 are associated with at least one blood phenotype. Similarly for lowlanders we observed 4 out of 7 SNPs that are associated with at least one blood phenotype.

In order to test whether these numbers of blood phenotype associations is larger/smaller than expected, we generated a background set by randomly sampling x windows (x being the number of candidate SNPs with at least one significant association in PNG highlanders or lowlanders) among the 200 bp windows associated with at least one phenotype in the UK biobank. We then recorded the number out of the x random windows that include a SNP associated with at least one blood phenotype. We resampled 10,000 sets of x windows, with x being equal to 5 or 7 in PNG highlanders and lowlanders, respectively. To test for significance, we computed a resampling P-value by calculating the proportion of random window sets in which the number of associations with at least one blood phenotype was higher than those observed windows associated with one blood phenotype. P-values were lower than 0.05 for both associated windows in PNG highlanders and lowlanders.

We are updating the methods and adding these results to the revised manuscript (page 10, line 307 - page 14, line 443)

3) Similarly, there's no formal test of whether there is an enrichment in archaic introgression or whether it's about what one would expect given the overall high levels of archaic introgression in these populations.

We agree with this reviewers' comment. In order to more accurately screen for the presence of archaic introgressed haplotypes in the selection candidate regions we used Skov HMM method on each PNG highlanders and lowlanders sample. (Skov et al., 2018), instead of using allele frequency for putative archaic SNPs in SGDP (page 2, line 62 – page 11, line 335, page 16, line 472 – Supplementary Tables 24-28).

In our new version of the manuscript, we describe the observation that several selection candidate regions contain archaic haplotypes and that some of the associated selection driver

SNPs are likely of Neandertal or Denisovan ancestry. In order to address the question on whether archaic haplotypes in selection candidate regions are over-represented it would require us to generate genome-wide introgression maps, an analysis that is not the primary scope of our study. We elaborated more on the need for further study in the revised manuscript (page 17, line 531).

In fact, a separate study that is currently ongoing and uses the same dataset to investigate the archaic ancestry in this cohort will investigate the patterns of selection on archaic haplotypes in more detail. The results of that study will be part of a different paper.

Skov, L., Hui, R., Shchur, V., Hobolth, A., Scally, A., Schierup, M. H., & Durbin, R. (2018). Detecting archaic introgression using an unadmixed outgroup. *PLOS Genetics*, 14(9), e1007641. <https://doi.org/10.1371/journal.pgen.1007641>

4) The authors do mention that associations in UK biobank might not transfer to the PNG populations but I think this is actually a pretty serious issue. How can we interpret these associations? The authors could look to see whether these specific associations replicate nominally in East Asian and African ancestry individuals from UKB.

We indeed highlighted that associations from the UK biobank have been detected in a different population than Papuans and that the transferability of the directionality of the beta values of the associations is, therefore, limited (Mathieson, 2021). Especially when the beta values are summed up to get polygenic score. Because of this, we avoided making any assumptions on the direction or the effect size of the phenotype association in PNG populations. However, most of the GWAS significant associations are due to common variants shared between populations or variants that map close to the associated SNPs (Wang et al., 2020). High replicability of GWAS results has notably been observed between Europeans and East Asians (Marigorta & Navarro, 2013). Thus, while we do not expect replicability of polygenic scores, we still expect the associated phenotype to remain the same between different populations. Moreover, we notice that UKBB summary statistics have already been used to infer polygenic selection in Papuan population (Choin et al., 2021). We clarified this in the revised manuscript (page 15, line 468).

Mathieson, I. (2021). The omnigenic model and polygenic prediction of complex traits. *The American Journal of Human Genetics*, 108(9), 1558-1563. <https://doi.org/10.1016/j.ajhg.2021.07.003>

Wang, Y., Guo, J., Ni, G., Yang, J., Visscher, P. M., & Yengo, L. (2020). Theoretical and empirical quantification of the accuracy of polygenic scores in ancestry divergent populations. *Nature Communications*, 11(1), 3865. <https://doi.org/10.1038/s41467-020-17719-y>

Marigorta, U. M., & Navarro, A. (2013). High Trans-ethnic Replicability of GWAS Results Implies Common Causal Variants. *PLOS Genetics*, 9(6), e1003566. <https://doi.org/10.1371/journal.pgen.1003566>

Choin, J. et al. Genomic insights into population history and biological adaptation in Oceania. *Nature* 1–7 (2021) doi:10.1038/s41586-021-03236-5.

5) is 1600-2400 M actually high enough to cause a substantial selective pressure related to altitude? The cited reference 8 says the following which suggest perhaps not:

““Although the expression ‘high altitude’ has no precise definition, the majority of individuals have certain clinical, physiological, anatomical and biochemical changes which occur at levels above 3000 m [~9,900 ft]. Individual variation is, however, considerable, and some people are affected at levels as low as 2000 m [~6,600 ft]” (West et al. 2007, p. 27; notes in brackets are added for clarification). Scientists often choose altitudes of 2,500 m (~8,250 ft) and above as a working definition of high altitude. “

We agree that the hypoxic pressure is lower in the PNG highlanders used in our study, than in Andeans and Tibetans usually studied for adaptation, to high-altitude. However, moderate hypoxia may still have a physiological impact. For example, rare occurrence of Acute Mountain

Sickness (AMS) and High-Altitude Pulmonary Edema (HAPE) (Luks et al., 2021). More importantly, reduced birth weight has been observed at 1,500 m a.s.l. in the United States (Yip, 1987) and shorter lifespan in adult Bolivians living above 1,500 m a.s.l. (Virues-Ortega et al., 2009).

Moreover, some populations, like the Andean Calchaquíes who have lived at intermediate altitude (above 1,500 m a.s.l and below 2,500 m a.s.l.) carry genomic signatures of selection to hypoxia (Eichstaedt et al., 2015). In addition, signature of positive selection to altitude has also been found among Ethiopians currently living at 1,800 m a.s.l. (Huerta-Sánchez et al., (2013)) and in Caucasus population living at intermediate altitudes of 2,000 m a.s.l. (Pagani et al.,2011). These studies suggest genomic signature of selection can occur even at intermediate altitude in response to moderate selection pressures.

We would also like to point out that physiological change immediately observed when ascending high or moderate high altitude might not be enough to truly estimate the strength of the selective pressure. For example, it has been observed that some mammals (including humans) tend to be smaller than their mainland counterpart when living at islands for a long time (i.e. Foster's rule or Island rule) (Foster, 1964). Physiological changes are not expected immediately after the mammals moved to an island or only after one generation. However, when adding up multiple generations for a long time, it might have enough effect to recognize through selection analysis.

If we agree that the selective pressure is milder in PNG highlands than at 4,000 m a.s.l. in Tibet, PNG highlands have been permanently for at least 20,000 years (Summerhayes, Field, Shaw, & Gaffney, 2017) or more than ten thousand of years earlier than Tibetans (Meyer et al., 2017) or Andeans (Rademaker et al., 2014). PNG highlanders have then been exposed much longer to this selective pressure. So if the selection acting on PNG highlanders is less strong than for Tibetans and Andeans, it had almost twice the time to drive changes in their genome, which might have more consequences.

Considering that the highlanders included in our study have lived at a similar altitudinal range (2300m, 2500m and 2700m), for thousands of years, we therefore believe that the hypothesis of selection to altitude in PNG highlanders remains valid.

We clarify these aspects in the introduction of revised manuscript (page 3, line 72 and 89)

Foster, J. B. (1964). Evolution of Mammals on Islands. *Nature*, 202(4929), 234-235.
<https://doi.org/10.1038/202234a0>

Summerhayes, G. R., Field, J. H., Shaw, B., & Gaffney, D. (2017). The archaeology of forest exploitation and change in the tropics during the Pleistocene : The case of Northern Sahul (Pleistocene New Guinea). *Quaternary International*, 448, 14-30.
<https://doi.org/10.1016/j.quaint.2016.04.023>

Meyer, M. C., Aldenderfer, M. S., Wang, Z., Hoffmann, D. L., Dahl, J. A., Degering, D., ... Schluetz, F. (2017). Permanent human occupation of the central Tibetan Plateau in the early Holocene. *Science*, 355(6320), 64-67. <https://doi.org/10.1126/science.aag0357>

Rademaker, K., Hodgins, G., Moore, K., Zarrillo, S., Miller, C., Bromley, G. R. M., ... Sandweiss, D. H. (2014). Paleoindian settlement of the high-altitude Peruvian Andes. *Science*, 346(6208), 466-469. <https://doi.org/10.1126/science.1258260>

6) Malaria seems like a much more plausible selective pressure but on the other hand it is already well-established for several decades that it has caused differential selection in these populations for example at globin genes – see Flint et al (1986) [<https://www.nature.com/articles/321744a0>] for example specifically in PNG. I'm surprised that this and related papers are not cited. Indeed, I wonder why the authors do not see signals of selection at Malaria related genes since as they themselves say, that is likely to be one of if not the strongest disease selection pressure.

We find that the GBP locus, under selection in PNG lowlanders is associated to malaria severity in the Cameroon population (Apinjoh et al., 2014). The paper of Flint et al. shows a correlation between the frequency of α^+ -thalassemia and malaria endemicity in PNG. α^+ -thalassemia would reduce severity of malaria which would make it a relevant candidate for selection studies in PNG. However, if α^+ -thalassemia is indeed under selection, it is more probably under balancing selection. If the heterozygotes only suffer from mild symptoms, homozygotes experience severe haemolytic anemia which requires regular blood transfusion and is a threat to survival. In addition, the more severe form α -thalassemia, when none of the 4 α -globin genes are expressed, leads to fetal death (Farashi & Hartevelde, 2018). Our selection scan design is unfortunately not able to detect balancing selection. We do not doubt that some of the peak (for example, GBP or SEC31A) are due to malaria (as it can differentially affect these populations), but we restrain ourselves from claiming that it is indeed caused by malaria as we do not have any direct evidence to prove that. We pointed out this limitation in the discussion (page 19, line 596).

Apinjoh, T. O., Anchang-Kimbi, J. K., Njua-Yafi, C., Ngwai, A. N., Mugri, R. N., Clark, T. G., ... in collaboration with The MalariaGEN Consortium. (2014). Association of candidate gene polymorphisms and TGF-beta/IL-10 levels with malaria in three regions of Cameroon : A case-control study. *Malaria Journal*, 13(1), 236. <https://doi.org/10.1186/1475-2875-13-236>

Farashi, S., & Hartevelde, C. L. (2018). Molecular basis of α -thalassemia. *Blood Cells, Molecules, and Diseases*, 70, 43-53. <https://doi.org/10.1016/j.bcmd.2017.09.004>

7) It is really hard to read the haplostrips figures in the main text supplement and see what the archaic haplotypes are. Perhaps a haplotype network would be easier to read and complement these figures.

We agree with this comment and thank the reviewer for this suggestion. In order to simplify visualization of shared haplotype between the archaic and PNG populations for the regions under selection, we built median-joining haplotype networks using popart ver. 1.7 (Leigh & Bryant, 2015) for 10kb windows centered on the candidate SNP when it was in high LD with the introgressed haplotype. New figures can be found in the revised manuscript and supplementary notes (page 12, line 363 – Supplementary Note 14– Figure 3 – Supplementary Figures 12-18).

Leigh, J. W., & Bryant, D. (2015). popart : Full-feature software for haplotype network construction. *Methods in Ecology and Evolution*, 6(9), 1110-1116. <https://doi.org/10.1111/2041-210X.12410>

8) Why use YRI as an outgroup for the PBS instead of say, Europeans who should also be an outgroup but would share the OOA bottleneck and be much closer in terms of genetic distance.

We chose to use Yorubans as an outgroup because we were interested in looking for potential introgressed haplotypes within the candidate regions for selection and we wanted to avoid masking adaptive introgressed regions common between PNG and CEU.

We clarify these aspects in the revised supplementary notes (page 8, line 232)

Typos etc

Line 420: "Archaic introgressions" should be "Archaic introgression"?

We thank the reviewer for noticing this typo edited the revised manuscript accordingly (page 17, line 510)

Line 451 "Cardio Vascular" should be "Cardiovascular phenotypes" or something?

We are grateful to the reviewer for noticing this typo edited the revised manuscript accordingly (page 17, line 533)

IV. Reviewer #2 (Remarks to the Author):

This study aims at detecting selection signals in two genetically closely related populations: highlanders and lowlanders from Papua New Guinea (PNG). The authors hypothesize that distinct environmental constraints have induced specific selective pressures on the two populations, which lead to unique genetic changes and phenotypic differences. The authors performed whole genome sequencing on 54 PNG highlanders and 74 PNG lowlanders. They performed a genome-wide selection scan and found several candidate genomic regions and SNPs under selection. The authors claim that many of these candidate SNPs play a role in gene expression or gene function and show association with cardiovascular-related and pathogenic-defense-related phenotypes in PNG highlanders and lowlanders, respectively. They also claim that many of the candidate genomic regions under selection harbor archaic introgression.

I think detecting selection signals in PNG highlanders and lowlanders is generally a very interesting and important topic. However, I have several major concerns about the analysis and conclusions reported in the study that need to be addressed before publication.

We thank the reviewer for reviewing the paper. We changed our manuscript according to the reviewer's comment and tried to address all the concerns.

1. Methods used for selection scan are invalid and could lead to false positives.

To infer the population structure of PNG individuals, the authors perform PCA and ADMIXTURE analysis using the merged dataset including newly sequenced PNG individuals, published Papuan genomes, and high-coverage genomes from the 1000 Genomes project from Africa (n=207), East Asia (n=202) and Europe (n=190). While this shows, PNG samples are very distinct from 1000 Genomes populations, they do not help much to characterize the genetic structure in PNG or their relationship to other groups. To better characterize the genetic ancestry of PNG individuals, samples from other populations like Oceanians from Human Genome Diversity Panel should be added. Both PCA and ADMIXTURE, however, show that PNG lowlanders in their dataset fall intermediate of PNG highlanders and PNG diversity panel and have ancestry from multiple groups when setting $K > 3$, suggesting admixture. However, the authors do not explore the demographic history of PNG highlanders and lowlanders. Admixture would have a major impact on the selection scan. Moreover, the PNG population is highly drifted (Brucato et al. 2021) which could also impact the power and false discovery rate of selection scans. The authors should perform formal tests of admixture, e.g., f-statistics, treeMix, etc. to explore the history of admixture and founder events in PNG groups. Importantly, if PNG lowlanders are admixed (as ADMIXTURE plots suggest), both PBS and XP-EHH could lead to false positives as the methods assume the test populations are homogenous. For example, the deviation of allele frequency detected in PBS by comparing PNG highlanders and PNG lowlanders could simply highlight regions of unknown ancestry in lowlands and not selection and the long haplotypes detected by XP-EHH may simply be due to signals of recent admixture. The authors need to first characterize the demographic history of the two populations and conduct selection scans with methods that account for admixture, like the method from Sanchez et al. (2013) (<https://doi.org/10.1093/molbev/mst089>).

We agree with the reviewer's comment and thanks the reviewer for suggesting to add different populations to the PCA and admixture displayed in this manuscript in order to better characterize the genetic structure in PNG. We now included 73 additional individuals to improve the resolution for these two analyses. These added individuals include two Australian genomes from SGDP (Mallick et al., 2016), 46 sequences from Flores Islands, in the East Island South-East Asia (East ISEA) (Jacobs et al., 2019), 10 samples from Mentawai islands and 15 from Nias Island, both in the West ISEA region (Jacobs et al., 2019). We edited the

revised manuscript at the relevant sections and included the updated figures (page 6, line 193 – Supplementary Note 1 – Supplementary Figures 3 and 4)

We performed selection scan only using PNG highlanders and PNG lowlanders samples. Both PNG highlanders and lowlanders population groups are relatively homogeneous and show little amount have very limited of admixture with East-Asians or ISEA, even when adding the additional Indonesian and Aboriginal Australian samples. On average, the admixture from East Asia and ISEA altogether account for 1% of PNG lowlanders ancestry at $k=8$ (i.e., the component with the lowest CV error). On the contrary, the PNG diversity dataset includes PNG samples originating from various places of the country (Bergström et al., 2020; Jacobs et al., 2019; Malaspinas et al., 2016; Mallick et al., 2016; Vernot et al., 2016) and display the important admixture present in these datasets. Although Papuan diversity set shows indeed substantial amount of admixture from East Asian ancestry, we have not used these samples for our selection analysis. We think the confusion might be due to low resolution of the graph, thus we have updated our Admixture analysis graph (Supplementary Figure 4).

Nonetheless, in order to explore further the extend of admixture in the PNG lowlanders sample we performed two Dstat tests. The first one with PNG highlanders, PNG lowlanders, CHB and YRI [Dstat(PNG HL, PNG LL, CHB, YRI)]. A second one with PNG highlanders, PNG diversity set I, CHB and YRI [Dstat(PNG HL, PNG diversity, CHB, YRI)] and a third one with PNG diversity set I, PNG lowlanders, CHB and YRI [Dstat(PNG diversity, PNG LL, CHB, YRI)]. The first test was not significant for any admixture ($Z = -0.145$) while the second ($Z = -22.874$), and third one ($Z = -30.339$), show significant admixture between the PNG diversity dataset I and CHB ($Z = -22.881$), which is supported by the ADMIXTURE analysis too. This supports that there is limited admixture in the PNG lowlanders samples that are used to perform the selection scans. We edited the revised manuscript and supplementary notes in order to include these results (page 7, line 214 - page 9, line 273 – page 13, line 385 - Supplementary Table 6).

We show that both PNG highlanders and lowlanders, the unique PNG populations used to perform selection scans show very limited admixture, in contrast with the PNG diversity sets PCA, admixture and D statistic results support that these two populations are homogeneous, show limited level of admixture and that admixture would have a minor impact on the selection scan.

Mallick, S., Li, H., Lipson, M., Mathieson, I., Gymrek, M., Racimo, F., ... Reich, D. (2016). The Simons Genome Diversity Project: 300 genomes from 142 diverse populations. *Nature*, 538(7624), 201-206. <https://doi.org/10.1038/nature18964>

Jacobs, G. S., Hudjashov, G., Saag, L., Kusuma, P., Darusallam, C. C., Lawson, D. J., ... Cox, M. P. (2019). Multiple Deeply Divergent Denisovan Ancestries in Papuans. *Cell*, 177(4), 1010-1021.e32. <https://doi.org/10.1016/j.cell.2019.02.035>

Bergström, A., McCarthy, S. A., Hui, R., Almarri, M. A., Ayub, Q., Danecek, P., ... Tyler-Smith, C. (2020). Insights into human genetic variation and population history from 929 diverse genomes. *Science*, 367(6484). <https://doi.org/10.1126/science.aay5012>

Malaspinas, A.-S., Westaway, M. C., Muller, C., Sousa, V. C., Lao, O., Alves, I., ... Willerslev, E. (2016). A genomic history of Aboriginal Australia. *Nature*, 538(7624), 207-214. <https://doi.org/10.1038/nature18299>

Vernot, B., Tucci, S., Kelso, J., Schraiber, J. G., Wolf, A. B., Gittelman, R. M., ... Akey, J. M. (2016). Excavating Neandertal and Denisovan DNA from the genomes of Melanesian individuals. *Science (New York, N. Y.)*, 352(6282), 235-239. <https://doi.org/10.1126/science.aad9416>

2. Sanity checks for selection scans needed.

Data QC: It is important that the authors include some quality metrics to show the reliability of their data (e.g., sample coverage, genotyping, and mapping quality, transition-transversion ratios, rates of heterozygosity, etc.). See 1000 Genomes and other genome coverage papers for commonly reported metrics. In addition, the authors should infer the phasing error rates and provide some evaluation of how phasing error, genotyping error, and variable coverage across samples impact their results. As shown in Figure S2, the samples used in this study

have an uneven distribution of coverage: PNG highlander samples have higher mean coverage than PNG lowlander samples. The authors should show that the selection signals are not biased by the uneven sample coverage across the two populations and that the selection signals are not limited to specific sets of samples of high/low coverage.

We added more quality metrics in addition to the mean of coverage (Figure S1), call rate and heterozygosity (Supplementary Figure 2, Supplementary Table 4, Supplementary Note 4). In the new supplementary table (Supplementary Table 5) we added nHets, nindels, average depth, nRefHom, nSNPs, rTiTv, rHetHom and rHet_NonRefHom computed for SNPs for all the individuals from our PNG samples and the different published datasets included in this paper. We observe that our PNG genomes remain in the variability already observed in published datasets.

In order to evaluate the phasing error, we computed the percentage of switch for individuals of the 1000G in our phased dataset that have phasing for trio available (CEU, n=1; YRI, n=2, KHV, n=2), which are regarded as more robust than statistical phasing. The average switch percentage in our phased dataset for these 5 individuals is of 1,41% which falls in the switch error range observed for the phased data published with the 1000 Genome Project for which the majority of SNPs, which fall in the MAF > 5% category, have an error < 2.5% (Belsare et al., 2019) (Supplementary Note 8).

We indeed have an uneven distribution of mean depth of coverage in our sample which might impact the detection of the selection signals. In order to check for the impact of including individuals with such higher mean coverage we decided to perform the XP-EHH and PBS selection scans on PNG highlanders and lowlanders while we have reduced the coverage for the 15 outliers (which are all PNG lowlanders) with mean of coverage >25 to 30% of their previous coverage (which roughly makes all the samples with very similar coverages).

In order to limit the influence of any other parameters, we performed four new independent variant calling, 2 used for the XP-EHH scans and 2 used for the PBS scans.

For XP-EHH, the first variant calling includes all the PNG highlanders and PNG lowlanders used in the initial XP-EHH selection scan. The second variant calling includes the same individuals with the 15 outlier lowlanders whose mean coverage has been reduced. We then followed the same phasing and XP-EHH selection protocol described in the manuscript. We see a huge correlation between the two XP-EHH scores (Spearman coeff=0.965) (Supplementary Figure 6a). Moreover, we plot the SNPs in the 1st percentile in the first XP-EHH scan (including high coverage lowlanders) and the SNPs in the 1st percentile in the second XP-EHH scan (with 15 outliers lowlanders with reduced coverage) (Supplementary Figure 6b,c). Most of our candidate region for selection in PNG lowlanders include SNP in the 1st percentile in the two XP-EHH scans.

In order to test the impact of higher coverage individual on the PBS scan, we performed two similar variant calling – with the 15 outliers with their initial mean of coverage or reduced coverage - including this time the YRI from 1000 Genomes that we use as an outgroup. We also observe a very strong correlation between PBS scores when using the high coverage lowlanders or the reduced coverage lowlanders (Spearman coeff=0.998) (Supplementary Figure 7).

These additional controls support that by having few individuals with higher coverage does not affect anyway to detect true selection region. This is most probably due to the choice of selection scan we have used in the manuscript. The top selected regions for both PBS and XPEHH mean shared SNP or SNP haplotypes between multiple individuals. As the variant calling was done on population level, the top region also means, a very high coverage for both reference and alternative alleles for total samples. Thus, the accuracy for those SNP or regions is high and unlikely affected by disproportionate coverage between individuals (at least for our tested scenario) or due to genotype error.

Overall, these additional analysis support that the PNG lowlanders with higher coverage have a limited impact on our selection scan results. We described further these additional analysis and result in the supplementary material (Supplementary Note 10).

Belsare, S., Levy-Sakin, M., Mostovoy, Y., Durinck, S., Chaudhuri, S., Xiao, M., ... Wall, J. D. (2019). Evaluating the quality of the 1000 genomes project data. *BMC Genomics*, 20. <https://doi.org/10.1186/s12864-019-5957-x>

Analysis QC: To detect genomic regions under selection, the authors first perform PBS on PNG highlanders and lowlanders setting YRI as an outgroup. They then perform XP-EHH and finally combine PBS and XP-EHH scores with Fisher score. For all these three steps, the authors keep regions with scores fall in 99th percentile and take the intercept of the three sets as the final candidate genomic regions. They then perform CLUES to find candidate SNPs driving selection in these regions. I find the pipeline for selection scan to be very puzzling: although the authors have combined PBS and XP-EHH results with Fisher score, they seem to still use 30 hits that are the top 10 in each criteria. Shouldn't they consider only the top hits based on Fisher score?

Moreover, the authors use Relate for inference of the ARGs. Recent studies have shown that Relate and other tree-based methods can have large uncertainty and errors in the inference of mutation ages (Brandt et al. 2022). Do the authors account this uncertainty in the CLUES analysis? How do these estimates impact their results? It would be useful if the authors can provide simulations to show that CLUES provides reliable results under their setup, especially for populations matching the demographic history of Papuans. Moreover, the authors need to at least include some sanity checks on their results: as positive controls, they should show if CLUES was able to detect well-known selection signals like LCT in EURs in their dataset and provide reasonable selection coefficient estimation. The authors should clearly state all parameters they input to CLUES. They should also provide results of Relate to show if the coalescent rates of the Papuan populations look reasonable. It would be useful if the authors can provide simulations to show that CLUES provides reliable results under their setup, especially for populations matching the demographic history of Papuans.

PBS and XP-EHH detect different types of selection signatures and their power also depends on the time of selection. They are correlated but not perfectly. PBS is based on allele frequency and can detect more ancient selective than XP-EHH that relies on haplotype homozygosity, which tends to be more recent. The Fisher Score only highlights the regions who show both these signature of selection (Cadzow et al., 2014). Thus, keeping only Fisher score is too conservative estimate (page 8, lines 227, page 9 - line 244).

We agree that we must clarify how we dealt with uncertainty and errors in the inference of mutation ages when generating the Relate output used by Clues, we edited the supplementary notes to clarify this (Supplementary Note 12). To produce the focal tree for each SNPs in the genomic regions of interest, we used the "SampleBranchLengths.sh" script that is part of Relate, with the "--num_samples 200" option. This will estimate the branch length (or mutation ages) from 200 cycles of resampling. In addition, with this resampling of the branch length, we also generated this focal tree 5 independent times. For each of the 5 focal generated by SNP, we ran CLUES independently. We then computed the average logLR from 5 runs for each SNP in a given genomic candidate region. The 5 SNPs with the highest average logLR value for a given genomic region were defined as top SNPs for the region. Additionally, we estimated focal tree and ran clues independently 50 times for the 5 top SNPs with the highest logLR average in the genomic region of interest and used the average log LR for the 50 runs as our selection estimate.

Similarly, we generated 5 independent focal trees for the CEU Relate trees and ran 5 independent runs for every SNPs within the MCM6/LCT locus (chr2: 135820191- 135876443).

We reproduce the results from the original Clues paper (Stern, Wilton, & Nielsen, 2019) by showing that the SNP with the highest logLR average is for rs4988235 – (chr2:135850976) (average logLR= 22.7596), a SNP of the MCM6 gene regulating the LCT and associated to the lactase persistence phenotype. We added this supplementary analysis to the supplementary notes (Supplementary Note 12). Suggesting that indeed our pipeline is capable of finding the candidate SNP.

Finally in order to support that our Relate setup is appropriate we added the Ne curves for CEU, CHB and YRI generated with our Relate set-up (Supplementary Figure 8). They are similar to the Ne curves generated in the original relate paper (Speidel, Forest, Shi, & Myers, 2019). None such curves were available for Papuan population or population with similar demographic history but the Ne curves we generated for PNG highlanders and lowlanders are similar to the Ne curves generated with MSMC2 (Schiffels & Wang, 2020) by Brucato et al. (Brucato et al., 2021) on other PNG individuals. Reproduction of these Ne curves supports that we have used an appropriate set up for Relate. (Supplementary Figure 8, Supplementary Note 11)

- Cadzow, M., Boocock, J., Nguyen, H. T., Wilcox, P., Merriman, T. R., & Black, M. A. (2014). A bioinformatics workflow for detecting signatures of selection in genomic data. *Frontiers in Genetics*, 5. <https://doi.org/10.3389/fgene.2014.00293>
- Stern, A. J., Wilton, P. R., & Nielsen, R. (2019). An approximate full-likelihood method for inferring selection and allele frequency trajectories from DNA sequence data. *PLOS Genetics*, 15(9), e1008384. <https://doi.org/10.1371/journal.pgen.1008384>
- Speidel, L., Forest, M., Shi, S., & Myers, S. R. (2019). A method for genome-wide genealogy estimation for thousands of samples. *Nature Genetics*, 51(9), 1321-1329. <https://doi.org/10.1038/s41588-019-0484-x>
- Schiffels, S., & Wang, K. (2020). MSMC and MSMC2 : The Multiple Sequentially Markovian Coalescent. In J. Y. Duthel (Éd.), *Statistical Population Genomics* (p. 147-166). New York, NY: Springer US. https://doi.org/10.1007/978-1-0716-0199-0_7
- Brucato, N., André, M., Tsang, R., Saag, L., Kariwiga, J., Sesuki, K., ... Ricaut, F.-X. (2021). Papua New Guinean Genomes Reveal the Complex Settlement of North Sahul. *Molecular Biology and Evolution*, (msab238). <https://doi.org/10.1093/molbev/msab238>

3. Rigorous association tests needed.

The authors use UK biobank summary statistics for EURs to look for significant phenotype association with candidate SNPs driving selection in their dataset. I think it is not a good idea to use genome-wide associations from UK biobank European ancestry since the authors have shown that PNGs have no European ancestry component in ADMIXTURE results and many studies have shown that GWAS results are not transferable across populations (as noted by the authors themselves). The authors instead use or at least show robustness to the use of genome-wide association statistics inferred from other populations such as Biobank Japan.

The authors also apply GEMMA to study the association between candidate SNPs and the phenotypes they measured, correcting for age, sex and height. Importantly, however, they do not correct for population structure even as they combine very diverse PNG lowlands and highland samples.

Moreover, as mentioned in lines 383-385, they did not find any significant phenotype association for top selection candidate SNPs when correcting for the number of SNPs and phenotypes tested together. This makes the result discussed in lines 361-379 puzzling and questionable since all of them are based on direct association tests of each SNP with the phenotype. The authors argue that this may be because of the low sample size, which might well be true but even without correction there are no significant hits and so this seems very speculative.

We indeed highlighted that associations from the UK biobank have been detected in a different population than Papuans. However, while we do not expect replicability of polygenic scores, we still expect the associated phenotype to remain the same between different populations. We clarified this in the revised manuscript (page 15, line 468). We also clarified this aspect in the answer to reviewer 1 4th comment.

In addition, UKBB includes healthy population-based prospective cohorts whereas BioBank Japan is a hospital-based disease-ascertained cohort. There are differences in observed heritability between these cohorts due to differences in phenotype precision, which would cause a lower prediction accuracy from the BBJ GWAS summary statistics for anthropometric and blood panel traits – that may be affected by the health condition and medication use of the patients - but higher prediction accuracy for five ascertained diseases. Regarding the hypothesis presented in our paper, we believe that the UKBB summary statistics are the most relevant to use because of their better accuracy for anthropometric and metabolic traits than the BBJ (Martin et al., 2019).

We performed the GEMMA analysis combining all the datasets we had phenotypes measurements available for (PNG diversity set I, PNG Highlanders, PNG lowlanders, n=234). PNG highlanders and lowlanders have different ancestry and the PNG diversity set I is very admixed. In order to correct for population stratification, we incorporated in GEMMA LMM a centered relatedness-matrix computed with GEMMA for the 234 samples included in this association analysis (page 11, line 330 – Supplementary Note 14).

We did find two significant association between two candidate SNPs and heart rate even when correction for the number of SNPs tested ($P_{\text{snp_adjusted}} < 0.05$). However, given the number of phenotypes we tested, these results did not remain significant after accounting for multiple testing. But the correction for this second criteria is quite complex regarding that most of these phenotypes are highly correlated (André et al. 2021). We agree with the reviewer's comment that the first sentence might be misleading regarding this challenge and edited the revised manuscript accordingly (page 15, line 448). Nonetheless, we believe that the results of association to multiple hematological phenotypes in the UKBB would support the suggestive association observed with heart rate. Given that to our knowledge, this kind of approach to use selective sweeps of special and underrepresented population to find the candidate SNP and then finding the associated phenotype is done for the first time. Although our approach did not survive the strict p-value test after using correction of multiple tests on phenotype, we still think this is very important to highlight, which will further encourage similar studies and develop this field in the near future.

Martin, A. R., Kanai, M., Kamatani, Y., Okada, Y., Neale, B. M., & Daly, M. J. (2019). Clinical use of current polygenic risk scores may exacerbate health disparities. *Nature Genetics*, 51(4), 584-591. <https://doi.org/10.1038/s41588-019-0379-x>

André, M., Brucato, N., Plutniak, S., Kariwiga, J., Muke, J., Morez, A., ... Ricaut, F.-X. (2021). Phenotypic differences between highlanders and lowlanders in Papua New Guinea. *PLOS ONE*, 16(7), e0253921. <https://doi.org/10.1371/journal.pone.0253921>

4. Inference of archaic ancestry.

The authors apply haplostrips to scan for regions with archaic haplotypes in candidate genomic regions under selection. This is not a rigorous way of looking for archaic ancestry in human genomes. To my understanding, haplostrips is a visualization method, no statistical tests are conducted on top of the clustering to show the actual probability of a genomic region coming from a specific ancestry. The authors should apply methods such as IBDmix (Chen et al. 2020), Sprime (Browning et al. 2018), HMM from Skov et al. (2018), DICAL-ADMIX (Steinrucken et al. 2018), etc. to infer archaic ancestry from the PNG genomes and conduct statistical tests to show that the regions under selection are enriched for haplotypes from archaic populations in populations of PNG individuals. As a sanity check, it would be useful if the authors can also

report the % of Neanderthal and Denisovan ancestry per individual in their study and use this information to assess if locally at their candidate loci there is significantly higher archaic ancestry.

We agree with this reviewers' comment. As further explained in our answer to reviewer 1 3rd comment, in order to better assess the presence of archaic introgressed variants in the regions candidates for selection we used Skov HMM method on each PNG highlanders and lowlanders sample with YRI as the unadmixed outgroup for the top regions for selections (Skov et al., 2018), instead of using allele frequency for archaic SNPs in SGDP (page 2, line 62 – page 11, line 335, page 16, line 472 – Supplementary Tables 24-28).

Skov, L., Hui, R., Shchur, V., Hobolth, A., Scally, A., Schierup, M. H., & Durbin, R. (2018). Detecting archaic introgression using an unadmixed outgroup. *PLOS Genetics*, 14(9), e1007641. <https://doi.org/10.1371/journal.pgen.1007641>

5. Quantitative assessment of signals is lacking.

There are many paragraphs in the manuscript where the authors discuss evidence based on raw count numbers without any statistical test of enrichment or significance. Some examples include line 393-402, 403-419. It is important to either provide clear tests of significance or remove these arguments.

We removed these arguments from the revised manuscript (page 16, line 489)

6. Release of data and scripts to ensure reproducibility.

Line 540-542 states that the authors would share their newly sequenced genomes upon publication. Would all the raw data and VCF files be deposited in EGA or other publicly accessible datasets? To ensure reproducibility of the main results, it is also important if the authors release all the data used for the analysis including the Papuan Diversity genomes (I tried to access this but this dataset is not available). Moreover, the authors should share their scripts for data processing and the selection scans (PBS, XP-EHH, CLUES, Relate, GEMMA, etc.) or clearly state all parameters and source files that were used. For example, it is unclear what human ancestor sequence did the authors use to polarize ancestral / derived alleles. And what parameters or filters were applied for various analysis.

The whole genome sequences (fastq and mapped cram files) as well as the phenotypes measurements from the 43 PNG highlanders and 38 lowlanders are available under restricted access to protect the privacy of the participants, in agreement with the Institutional Review Board approval and the individuals informed consents forms. The whole genome sequences have been deposited in the European Genome-phenome Archive (<https://www.ebi.ac.uk/ega/>) with the accession number: EGAD00001010143 (<https://ega-archive.org/datasets/EGAD00001010143>). The data are available to the scientific community through EGA, under controlled access review by the Data Access Committee of the Papua New Guinean Genome Diversity Project (contact person: francois-xavier.ricaut@univ-tlse3.fr)

The sequences included in the Papuan Diversity set I are accessible there : <https://ega-archive.org/studies/EGAS00001005393> but are under controlled access review by the Data Access Committee of the Papua New Guinean Genome Diversity Project.

Our custom scripts are available on GitHub: <https://github.com/mathilde999/selection-png>

We edited the revised manuscript accordingly (page 5, line 154 – page 21, line 627)

Minor points:

1. The authors found 21 regions after selection scan. However, they don't keep the same set of candidate regions in all following analyses. For example, in the section on looking for archaic segments they use 44 genomic regions instead of 21.

After selection scan we defined 21 regions under selection in PNG highlanders but also 23 regions under selection in PNG lowlanders. We then performed the following analyses, such as the section looking for archaic segments, on the regions under selection in PNG highlanders (n=21) and the regions under selection (n=23) or 44 genomic regions in total. We agree that grouping the different regions under selection in the two populations was unclear and we edited this section for clarity (page 2, line 60 – page 17, line 511).

2. In Supplementary Note 1c, the authors use (The 1000 Genomes Project Consortium et al., 2015) as the reference for their 1000 Genome high coverage data. This is confusing since this paper only releases low coverage Phase III 1000 Genome data based on my understanding. They should probably cite (<https://doi.org/10.1016/j.cell.2022.08.004>) instead.

It is indeed the high Coverage paper as the reviewer suggested. We edited the manuscript accordingly (page 7, lines 165,190).

3. Typo “hypothesise” in the abstract.

We thank the reviewer to find the typo and edited the manuscript accordingly (page 2, line 43)

4. Line 136 - sampled between 2016 and 2019 (EGA accession code XXXXX). Update EGA accession number.

We update the manuscript accordingly (page 6, line 169)

5. Line 440-441: This fact and the gene flow between the Altai Neanderthal and Denisova would suggest that we most likely observed Denisovan introgression within the GBP locus in the PNG population. Is there evidence that Altai Neanderthal has Denisova ancestry in this region?

In order to see if there is any evidence of shared ancestry between Altai Neanderthal and Denisova for the candidate region chr1:88800562-89326878 including the GBP locus, we performed a pairwise comparison of the distance between the high coverage genomes of three Neanderthals and the Denisovan for the introgressed haplotype in this region. We observed that the Altai Neanderthal and Denisovan showed the closest sequence similarity. They are more than twice as distant as comparison of to any of the other two Neanderthals (Vindija, Chagyrskaya). These results suggest that, indeed, this region might be a candidate for admixture between the Altai Neanderthal and the Denisovan. We included this additional analysis to the revised manuscript (page 17 , line 520 – Supplementary Table 28).

REVIEWER COMMENTS

Reviewer #1 (Remarks to the Author):

I appreciate the extra work that the authors have put into this manuscript. I think the approaches they have taken to assess significance of their selection scan results are reasonable. However, I feel that the main text hasn't been fully updated to reflect that. When I read the main text, I came away with a very different impression than when I started to look at the supplement. For example, the text talks about considering anything with $P < 0.05$ (approximately $|Z| > 2.8$) as significant and the response talks about anything with $|Z| > 3$ as significant but these are nowhere near genome-wide significance levels, so I assumed that nothing was significant. But then when I looked in the supplement, I saw that the reported Z scores were all much larger than that.

Overall, the manuscript is improved and some of the selection scan results look significant. But I think they still need some work on their presentation.

The claims about phenotypic associations are well-caveated in the text (you certainly need to correct for both the number of SNPs and the number of phenotypes), but perhaps a bit oversold in the abstract. Unfortunately, the study is just too underpowered for this analysis.

Finally, I also feel that the claims about archaic admixture should be removed, and some aspects of the data origin and access need to be clarified.

1) Presentation of results: Z scores for the reported loci should be given in Tables 1 and 2. I think the Manhattan plots in Figure 1 would be better with $(-\log_{10})$ P-values for the XP-EHH and PBS analyses, or with Z scores for everything so that they are comparable across plots. You should show the lowlander XP-EHH scores. Each Manhattan plot should have a qq-plot so we can see how well calibrated the P-values are. If they are not, consider you should apply a correction like genomic control to the P-values.

2) Unless I'm missing something, the archaic ancestry analysis only tells you whether there are any archaic haplotypes in each region – in general there's no evidence that the individual archaic haplotypes are under selection, so not really adaptive introgression (except maybe GBP?). As the authors say, without a genome-wide map of introgression, they cannot really assess whether there is more archaic admixture at candidate selected loci, so the claim that archaic admixture is important (last sentence of the abstract) should be removed.

3) I think it should be made clear what (if any) the conditions for data access are. Will the author make the data available for only specific types of analysis, is there any agreement needed etc? – please give details of the required agreement. Summary selection scan results should be made openly available in full in the supplementary information or some permanent repository like zenodo (i.e. per-snp PBS and XP-EHH results). These are the main product of the paper.

4) Some of the data are described as "personal communications" by the senior author (line 160 and Table S1). I don't think that's acceptable – these data should be part of the paper and included in the data release. Important data as personal communication is not appropriate, especially when it's a communication from one of the authors! Similarly, I don't think you can acknowledge the senior author for providing data (line 658). That's why they are an author! (unless this refers to a different F.-X. Ricaut, but even then it's still not appropriate to be a personal communication).

Minor comments:

5) Some of the UK Biobank associations reported in supplementary tables S9 and 10 are embarrassing. "Canned Soup intake"? "Place of birth in UK"? Those are not real or relevant associations. I think you should apply some kind of significance cutoff here otherwise it just looks silly.

6) In general, I still feel like there are a lot of typos, though I haven't gone systematically through the manuscript:

Line 169: "described".

Line 170: "Australians"

Line 209: Cross-validation in ADMIXTURE does not tell you "the most likely number of components". There is no such thing.

Line 260: Missing parenthesis

Line 458: "adjusted_sno"

Line 446: "haematological" [phenotypes?]

Line 448: "phenotypes measurements done for"... [phenotype measurements for?]

Reviewer #2 (Remarks to the Author):

The authors properly addressed some of the technical concerns that have been raised. However, I still have reservations about some results and conclusions that I describe below.

The authors claim that the PCA, ADMIXTURE and D statistic results support that the two PNG populations are relatively homogeneous and show limited levels of admixture from East Asia and ISEA (FigS3, S4, Table S6). However, the Dstatistics they apply only test for admixture from CHB into Papuan LL or HL and so do not investigate recent gene flow between Papuan LL and HL and Papuan LL and Papuan Diversity set, the main concern that I raised earlier. Note, Admixture $K < 8$ suggests Papuan LL might be an admixture between Papuan HL and Papuan Diversity set – though Admixture results are very sensitive to drift and so should not be used as formal tests (Fig S4, $k = 5-7$ in black). It would be useful if the authors apply the following Dstatistics or f-statistics -- $D(\text{outgroup}, \text{PNG LL}; \text{PNG diversity}, \text{YRI})$ and $D(\text{outgroup}, \text{PNG LL}; \text{PNG HL}, \text{YRI})$ where outgroup can be CEU, or apply Treemix. If Papuan LL is admixed with HL, it is not valid to use them as a reference population in PBS. Also, XP-EHH assumes the population is homogenous and for admixture populations, it can give inaccurate results, as highlighted earlier. Thus, authors should apply other methods that are robust to admixture in the target populations such as Sanchez et al. (2013) (<https://doi.org/10.1093/molbev/mst089>).

The authors have addressed the rest of my comments. I found the new analysis and figures added for data quality metrics and their impact on selection scans to be very helpful. Moreover, additional details and sanity checks on Relate and CLUES analysis make the results more convincing. And the archaic ancestry inference with Skov HMM (2018) does show the interesting potential archaic ancestry on GBP locus in PNG lowlanders. I also appreciate the authors' effort to share the data and analysis pipeline to make the results reproducible.

REVIEWER COMMENTS

Reviewer #1 (Remarks to the Author):

I appreciate the extra work that the authors have put into this manuscript. I think the approaches they have taken to assess significance of their selection scan results are reasonable. However, I feel that the main text hasn't been fully updated to reflect that. When I read the main text, I came away with a very different impression than when I started to look at the supplement. For example, the text talks about considering anything with $P < 0.05$ (approximately $|Z| > 2.8$) as significant and the response talks about anything with $|Z| > 3$ as significant but these are nowhere near genome-wide significance levels, so I assumed that nothing was significant. But then when I looked in the supplement, I saw that the reported Z scores were all much larger than that.

Overall, the manuscript is improved and some of the selection scan results look significant. But I think they still need some work on their presentation.

The claims about phenotypic associations are well-caveated in the text (you certainly need to correct for both the number of SNPs and the number of phenotypes), but perhaps a bit oversold in the abstract. Unfortunately, the study is just too underpowered for this analysis.

Finally, I also feel that the claims about archaic admixture should be removed, and some aspects of the data origin and access need to be clarified.

We thank the reviewers for their valuable comments. We updated the abstract and the main text to reflect the suggestions above. We have downplayed the claim regarding the association with heart rate in the abstract (page 2, line 55). We addressed the other comments below.

1) Presentation of results: Z scores for the reported loci should be given in Tables 1 and 2. I think the Manhattan plots in Figure 1 would be better with $(-\log_{10})$ P-values for the XP-EHH and PBS analyses, or with Z scores for everything so that they are comparable across plots. You should show the lowlander XP-EHH scores. Each Manhattan plot should have a qq-plot so we can see how well calibrated the P-values are. If they are not, consider you should apply a correction like genomic control to the P-values.

We thank the reviewers for this wonderful suggestion. We recomputed the Z score and p-value after \log_{10} transformation of PBS and Fisher scores in order to make their distribution normal after looking at the qqplot (Supplementary figure 7). The new Z score and p-value were updated in the supplementary tables S7 and S8. We also added the qqplot for the scores (XP-EHH, $\log_{10}(\text{PBS})$ and $\log_{10}(\text{Fisher score})$) and the Manhattan plot with the p-value as supplementary figures (Supplementary figures 5 and 6). We have split the previous main figure 1 into two figures (Figures 1 and 2), one displaying Manhattan plots for PNG highlanders and the second showing the Manhattan plots for PNG lowlanders.

2) Unless I'm missing something, the archaic ancestry analysis only tells you whether there are any archaic haplotypes in each region – in general there's no evidence that the individual archaic haplotypes are under selection, so not really adaptive introgression (except maybe GBP?). As the authors say, without a genome-wide map of introgression, they cannot really assess whether there is more archaic admixture at candidate selected loci, so the claim that archaic admixture is important (last sentence of the abstract) should be removed.

We have updated the abstract accordingly to lower the claim for the importance of adaptive introgression in the abstract (page 2, line 55)

3) I think it should be made clear what (if any) the conditions for data access are. Will the author make the data available for only specific types of analysis, is there any agreement needed etc? – please give details of the required agreement. Summary selection scan results should be made openly available in full in the supplementary information or some permanent repository like zenodo (i.e. per-snp PBS and XP-EHH results). These are the main product of the paper.

We thank the reviewer for their comment and updated the data availability section to display the condition for data access in the reviewed manuscript (p21, line 599). We have deposited the per-SNP PBS and XP-EHH results (as well as the CLUES per-SNP log(LR) and hmix frequency) on fig share, where they will be made publicly available after publication. We have updated the Data availability accordingly (page 22, line 637). Reviewers can already access them via this private review link: <https://figshare.com/s/256b55c043be66b62649>.

4) Some of the data are described as "personal communications" by the senior author (line 160 and Table S1). I don't think that's acceptable – these data should be part of the paper and included in the data release. Important data as personal communication is not appropriate, especially when it's a communication from one of the authors! Similarly, I don't think you can acknowledge the senior author for providing data (line 658). That's why they are an author! (unless this refers to a different F.-X. Ricaut, but even then it's still not appropriate to be a personal communication).

We have now deposited all of the new whole genome data on EGA under three different datasets:

- EGAD00001010143: Papua New Guinean Genome Altitude Project Dataset 1: The PGAP dataset 1 includes 81 whole genome sequences for Papua New Guinean individuals sampled in Daru (N=38) and Mount Wilhelm (N=43)
- EGAD00001010142: Papua New Guinean Genome Altitude Project Dataset 2: includes 82 whole genome sequences for Papua New Guinean individuals sampled in Daru (N=1), Port Moresby (N=64) and Mount Wilhelm (N=17).
- EGAD50000000050: Papua New Guinean Lowlanders Dataset: includes 41 whole genome sequences for Papua New Guinean individuals sampled in Daru.

We updated the reviewed manuscript (page 14, line 385 and page 22, line 630) and the supplementary (Table S1 and Supplementary Note S1) accordingly.

Minor comments:

5) Some of the UK Biobank associations reported in supplementary tables S9 and 10 are embarrassing. "Canned Soup intake"? "Place of birth in UK"? Those are not real or relevant associations. I think you should apply some kind of significance cutoff here otherwise it just looks silly.

We thank the reviewer for their comment. Following their recommendation, we excluded from the phenotypes related to diet (categories 100090 and 100052), sociodemographic factors (category 100062), sexual factors (category 100056), electronic use (category 100053), and the local environment (category 114) because of the absence of correspondence between the environmental variable described within these categories and the environment to which the studied PNG populations are exposed. This has led to a change in the p-value significance threshold for the UK Biobank association section. We updated the main manuscript (page 18,

line 537 and Tables 1 and 2) and the supplementary material (Supplementary Note S13 and tables S13-S15) accordingly.

6) In general, I still feel like there are a lot of typos, though I haven't gone systematically through the manuscript:

We thank the reviewer for flagging these typos. We carefully reviewed the manuscript and addressed the typos below and the remaining ones.

Line 169: "described".

Updated

Line 170: "Australians"

Done

Line 209: Cross-validation in ADMIXTURE does not tell you "the most likely number of components". There is no such thing.

We have updated this sentence in both the main manuscript (page 16, line 450) and supplementary (Supplementary Note S7 and supplementary figure S4).

Line 260: Missing parenthesis

Corrected

Line 458: "adjusted_sno"

Corrected

Line 446: "haematological" [phenotypes?]

Corrected

Line 448: "phenotypes measurements done for"... [phenotype measurements for?]

Corrected

Reviewer #2 (Remarks to the Author):

The authors properly addressed some of the technical concerns that have been raised. However, I still have reservations about some results and conclusions that I describe below.

We thank the reviewers for their comments. We addressed the reservation mentioned by the reviewer below.

The authors claim that the PCA, ADMIXTURE and D statistic results support that the two PNG populations are relatively homogeneous and show limited levels of admixture from East Asia and ISEA (FigS3, S4, Table S6). However, the Dstatistics they apply only test for admixture from CHB into Papuan LL or HL and so do not investigate recent gene flow between Papuan LL and HL and Papuan LL and Papuan Diversity set, the main concern that I raised earlier. Note, Admixture $K < 8$ suggests Papuan LL might be an admixture between Papuan HL and Papuan Diversity set – though Admixture results are very sensitive to drift and so should not be used as formal tests (Fig S4, $k = 5-7$ in black). It would be useful if the authors apply the following Dstatistics or f-statistics -- $D(\text{outgroup}, \text{PNG LL}; \text{PNG diversity}, \text{YRI})$ and $D(\text{outgroup}, \text{PNG LL}; \text{PNG HL}, \text{YRI})$ where outgroup can be CEU, or apply Treemix. If Papuan LL is admixed with HL, it is not valid to use them as a reference population in PBS. Also, XP-EHH assumes the population is homogenous and for admixture

populations, it can give inaccurate results, as highlighted earlier. Thus, authors should apply other methods that are robust to admixture in the target populations such as Sanchez et al. (2013) (<https://doi.org/10.1093/molbev/mst089>).

Indeed, as the reviewer pointed out, some PNG lowlanders show admixture from PNG highlanders. Although the level of admixture from highlanders to lowlanders is low on average (3.23% with a standard deviation of $\pm 5.29\%$), some lowlanders do show a substantial amount of highlander ancestry. We also have worked previously on admixed populations and demonstrated that major improvements could be gained, particularly on PBS analysis (Yelmen et al 2021 GBE), by using a masking strategy. We did not use this strategy here for two reasons. First, the amount of admixture is much lesser (less than 5 %), and second, the component in both the cases for Yelmen et al and Huerta-Sánchez et al. is coming from Europeans admixed with South Asians and Ethiopians, respectively, which might have created major shifts in the selection analysis. In our particular case, the admixture occurred between two Papuan populations. Based on the last few decades of population genetics analysis on human populations, we think that finding a population without contribution from neighbouring populations is nearly impossible as a few migrations between close populations always exist.

In our particular case, given that neighbouring highlanders contributed to lowlanders, using highlanders as a reference population in both XPEHH and PBS will mask strong signatures of selection coming from highlanders within lowlanders. In other words, because we used a cross-population approach, admixed regions within lowlanders will also be present in highlanders - in the case of XPEHH, the same haplotype will be found in both highlanders and lowlanders and in the case of PBS allele frequency of those regions will be similar between these two populations – and such genomic regions will not have as strong values as signatures of selection only present in lowlanders but absent in highlanders (in another word unlikely to be the top candidates). Nonetheless, to prove our point, we removed lowlanders with more than 5% highlander ancestry and our selection scan analysis again (Supplementary Figures 8 and 9 and Supplementary Note 11). We found that the results are highly correlated (Spearman coefficient > 92%) with the results from the initial analysis, and most of our target regions reproduced in the unadmixed samples, suggesting PNG highlander contribution in PNG lowlander does not particularly affect our analysis.

However, we agree that this is a caveat to our study design that aimed to detect regions under selection unique to PNG highlanders or PNG lowlanders and necessitate close populations between which we can't avoid residual admixture. We mentioned this in the reviewed manuscript (page 6, line 136).

Yelmen, B. et al. Improving Selection Detection with Population Branch Statistic on Admixed Populations. *Genome Biology and Evolution* 13, evab039 (2021).

Huerta-Sánchez, E. et al. Genetic Signatures Reveal High-Altitude Adaptation in a Set of Ethiopian Populations. *Mol Biol Evol* 30, 1877–1888 (2013).

The authors have addressed the rest of my comments. I found the new analysis and figures added for data quality metrics and their impact on selection scans to be very helpful. Moreover, additional details and sanity checks on Relate and CLUES analysis make the results more convincing. And the archaic ancestry inference with Skov HMM (2018) does show the interesting potential archaic ancestry on GBP locus in PNG lowlanders. I also appreciate the authors' effort to share the data and analysis pipeline to make the results reproducible.

We thank the reviewer for their valuable comments.

REVIEWERS' COMMENTS

Reviewer #1 (Remarks to the Author):

The authors have addressed all my comments from the previous round. I have no further suggestions.

Reviewer #2 (Remarks to the Author):

The authors have addressed my concerns. Below are a few final minor comments:

Line 112: Tibetans – Tibetans`

Line 183: for all the SNPs – for all SNPs

Line 187: each candidate SNPs – each candidate SNP

Line 327: shows

Line 362: an association – associations

REVIEWERS' COMMENTS

Reviewer #1 (Remarks to the Author):

The authors have addressed all my comments from the previous round. I have no further suggestions.

We thank reviewer 1 for their thorough review across the different rounds of review.

Reviewer #2 (Remarks to the Author):

The authors have addressed my concerns. Below are a few final minor comments:

We thank reviewer 2 for their thorough review across the different rounds of review. We updated the review manuscript to include those minor comments; see below.

Line 112: Tibetans – Tibetans`

Corrected – page 4, line 107

Line 183: for all the SNPs – for all SNPs

Corrected – page 7, line 177

Line 187: each candidate SNPs – each candidate SNP

Corrected – page 7, line 183

Line 327: shows

Corrected – page 12, 322

Line 362: an association – associations

Corrected – page 13, 358